# CREPE: 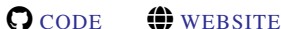
# CONTROLLING DIFFUSION WITH REPLICA EXCHANGE

**Jiajun He**[1,*]  **Paul Jeha**[2,†]  **Peter Potaptchik**[3,†]  **Leo Zhang**[3,†]

**José Miguel Hernández-Lobato**[1]  **Yuanqi Du**[4]  **Saifuddin Syed**[5,‡]  **Francisco Vargas**[6,‡]

[*]Corresponding to jh2383@cam.ac.uk. [†]Equal contributions. [‡]Last authors.

[1]University of Cambridge, [2]Technical University of Denmark, [3]University of Oxford,
[4]Cornell University, [5]University of British Columbia, [6]Xaira Therapeutics.

 CODE    WEBSITE

## ABSTRACT

Inference-time control of diffusion models aims to steer model outputs to satisfy new constraints without retraining. Previous approaches have mostly relied on heuristic guidance or have been coupled with Sequential Monte Carlo (SMC) for bias correction. In this paper, we propose a flexible alternative based on replica exchange, an algorithm designed initially for sampling problems. We refer to this method as CREPE (Controlling with REPlica Exchange). Unlike SMC, CREPE: (1) generates particles sequentially, (2) maintains high diversity in the generated samples after a burn-in period, and (3) enables online refinement or early termination. We demonstrate its versatility across various tasks, including temperature annealing, reward-tilting, model composition and classifier-free guidance debiasing, with competitive performance compared to prior SMC methods.

## 1 INTRODUCTION

Diffusion models (Ho et al., 2020; Song et al., 2021b;a) have revolutionised generative modelling with their ability to produce high-quality samples across diverse modalities, including images (Rombach et al., 2022; Karras et al., 2022), videos (Ho et al., 2022), text (Austin et al., 2021), among others (Watson et al., 2023; Duan et al., 2023). It is typically formalised as a stochastic process initialised at a tractable distribution (e.g., a Gaussian distribution or a fully masked distribution) and evolves to recover the data distribution. This progressive nature not only enables diffusion models to excel at modelling complex distributions but also provides flexible approaches for steering the generation.

*Inference-time control* leverages this flexibility to steer the generation of diffusion models, enabling tasks such as posterior sampling (Dou & Song, 2024), reward-tilting (Wu et al., 2023; Singhal et al., 2025), tempering (Akhound-Sadegh et al., 2025), or model composition (Du et al., 2023). This

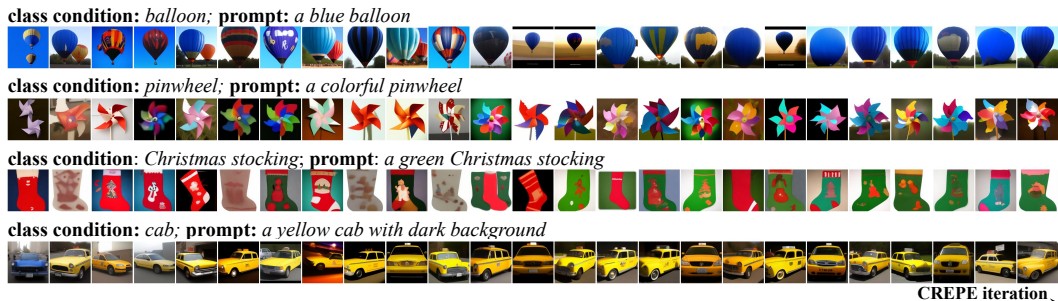

Figure 1: Trajectory of images generated using CREPE for prompted reward-tilting on ImageNet-512, thinned every 8 iterations. After burn-in, the samples align closely with the prompt.

Figure 2: Comparison between diffusion inference-time control with SMC and CREPE. We visualise the diffusion process using colour shading: darker colours correspond to higher noise/mask levels (large $t$), while brighter colours indicate states closer to the data distribution (small $t$). **SMC (Left):** particles are initialised at $t = 1$ and progressively denoised towards lower noise levels. During denoising, importance resampling is applied to select particles that better satisfy the imposed constraints. **CREPE (Right):** particles are initialised at several different diffusion steps, and they undergo local exploration and communication in parallel, evolving them towards desired constraints. The example shows using SMC and CREPE to debias classifier-free guidance.

was first explored through classifier (and classifier-free) guidance (Dhariwal & Nichol, 2021; Ho & Salimans, 2022), and has since been extended with a variety of approximation or fine-tuning approaches (Song et al., 2023a; Chung et al., 2023; Song et al., 2023b; Schneuing et al., 2024; Ye et al., 2024; Kong et al., 2025; Denker et al., 2024; Liu et al., 2024; Domingo-Enrich et al., 2024; Zhang et al., 2024). However, these methods often rely on heuristic approximations and typically suffer from inaccuracies, while fine-tuning approaches require additional training on data and may still suffer from imperfect optimisation. This has motivated studies to debias the errors (Wu et al., 2023; Skreta et al., 2025; Lee et al., 2025; Singhal et al., 2025; Thornton et al., 2025).

A commonly used framework to debias is Sequential Monte Carlo (SMC, Del Moral et al., 2006), where one jointly evolves a set of weighted interacting particles along the generation path towards the desired distribution. SMC debiases the trajectories by importance sampling with potential resampling moves. However, SMC-based debiasing methods typically suffer from several limitations: (1) it requires maintaining a large number of particles throughout the denoising trajectory, which can be memory-intensive; (2) SMC tends to suffer from poor sample diversity, as observed in several recent works (Li et al., 2024; Young & Akyildiz, 2024; Lee et al., 2025), a problem that is especially severe when the number of particles is small; (3) once the sampling process is complete, SMC cannot refine the generated samples. If the outcome is unsatisfactory or new constraints are added, one needs to regenerate new samples rather than iterate on the existing ones.

On the other hand, replica exchange, also known as Parallel Tempering (PT, Swendsen & Wang, 1986; Geyer, 1991; Hukushima & Nemoto, 1996), is a Markov Chain Monte Carlo (MCMC) algorithm providing a computationally dual framework to SMC. PT reverses the roles of parallelism and time in SMC samplers (Syed et al., 2024): instead of propagating a batch of particles in parallel along the denoising direction sequentially, PT runs a chain at different denoising steps in parallel and generates particles sequentially. This reduces the burden of parallelism over a large number of particles and enables the continual refinement of the generated samples. However, standard PT and its extensions (Ballard & Jarzynski, 2009; Zhang et al., 2025) were designed for sampling from unnormalised densities, where the target distribution is explicitly known. This highlights a key challenge: *in inference-time control, we only have access to a pretrained diffusion model. Can PT still be adapted to this setting to harness its desirable properties?*

In the following, we answer this question affirmatively. In summary:

- We formulate inference-time control with parallel tempering (PT) for diffusion models. Dubbed as Control with REPlica Exchange (CREPE), it shows how PT can be applied directly from pretrained diffusion models without explicit target densities.

- We derive PT swap rates for several inference-time control applications, including tempering, reward tilting, debiasing classifier-free guidance, and model composition, for both the Gaussian diffusion model and discrete mask diffusions (Lou et al., 2023).

- We validate our approach across various tasks and modalities, demonstrating improved performance and better inference-time scaling property compared to SMC-based approaches.

## 2 BACKGROUND

We begin with a discussion of path measures for continuous-time Markov processes, followed by an introduction to diffusion models and their discrete counterparts (Song et al., 2021b; Lou et al., 2023; Shi et al., 2023). Finally, we review replica exchange, aka parallel tempering (PT), particularly focusing on its accelerated variants (APT, Zhang et al., 2025).

### 2.1 PATH RADON-NIKODYM DERIVATIVE AND RADON-NIKODYM ESTIMATOR

Let $\overrightarrow{X}_s$ and $\overleftarrow{X}'_s$ be continuous-time Markov process on state space $\mathbb{X}$ within the time interval $s \in [t, t']$. In later sections, we will instantiate these processes as either diffusion or jump processes; for now, we keep the discussion general. Let $\overrightarrow{\mathbb{F}}_{t,t'}$ denote the path measures defined as the law of the forward process $\overrightarrow{X}_{[t,t']} = (\overrightarrow{X}_s)_{s \in [t,t']}$ obtained by evolving $\overrightarrow{X}_t \sim \mu$ forward from time $t$ to $t'$. Similarly let $\overleftarrow{\mathbb{B}}_{t,t'}$ denote the path measure for the backward process $\overleftarrow{X}'_{[t,t']} = (\overleftarrow{X}'_s)_{s \in [t,t']}$ obtained by evolving $\overleftarrow{X}_{t'} \sim \mu'$ backward from time $t'$ to time $t$. For a more intuitive introduction, let's first consider the forward and backward processes at a given collection of time points $t = t_0 < t_1 < \cdots < t_K = t'$. The laws can be factored in terms of the forward transition kernels $\overrightarrow{F}_{t_k|t_{k-1}}$ of the forward process and the backward transition kernels $\overleftarrow{B}_{t_{k-1}|t_k}$ of the backward process, i.e.,

$$\overrightarrow{\mathbb{F}}_{t,t'}(x_{t_{0:K}}) = \mu(x_{t_0}) \prod_{k=1}^{K} \overrightarrow{F}_{t_k|t_{k-1}}(x_{t_k}|x_{t_{k-1}}), \overleftarrow{\mathbb{B}}_{t,t'}(x'_{t_{0:K}}) = \mu'(x'_{t_K}) \prod_{k=1}^{K} \overleftarrow{B}_{t_{k-1}|t_k}(x'_{t_{k-1}}|x'_{t_k}). \quad (1)$$

By taking ratios and a formal limit as $\max_k |t_k - t_{k-1}| \to 0$ (Berner et al., 2025, Proposition B.7.), we obtain the Radon-Nikodym derivative between $\overrightarrow{\mathbb{F}}_{t,t'}$ and $\overleftarrow{\mathbb{B}}_{t,t'}$ in the form of the density ratio between the marginals $\mu$ and $\mu'$, and a term $R_{t,t'}$ (Vargas et al., 2024; Berner et al., 2025).

$$\frac{d\overleftarrow{\mathbb{B}}_{t,t'}}{d\overrightarrow{\mathbb{F}}_{t,t'}}(x_{[t,t']}) = \frac{\mu'_{t'}(x_{t'})}{\mu_t(x_t)} R_{t,t'}(x_{[t,t']}). \quad (2)$$

Formally, $R_{t,t'}(x_{[t,t']})$ is defined as the ratio of the backward transition dynamics initialised terminating at $x_{t'}$ and the forward transition dynamics initialised at $x_t$ in the limit as $\max_k |t_k - t_{k-1}| \to 0$,

$$R_{t,t'}(x_{[t,t']}) = \lim_{\max_k |t_{k+1} - t_k| \to 0} \frac{\prod_{k=1}^{K} \overleftarrow{B}_{t_{k-1}|t_k}(x_{t_{k-1}}|x_{t_k})}{\prod_{k=1}^{K} \overrightarrow{F}_{t_k|t_{k-1}}(x_{t_k}|x_{t_{k-1}})}. \quad (3)$$

We note that $R_{t,t'}$ depends only on the transition dynamics of forward and backward processes, independent of the marginals $\mu$ and $\mu'$. When $X_t$ and $X'_t$ are constructed via a stochastic differential equation (SDE) or continuous-time Markov chain (CTMC), we can express $R_{t,t'}(x_{[t,t']})$ analytically in terms of a path integral of the drift and rate matrices respectively over $x_{[t,t]}$, which we describe in Appendix A.2. Going forward we will refer to $R_{t,t'}$ as the Radon-Nikodym Estimator (RNE) between the forward and backward process over the interval $[t, t']$ following He et al. (2025c).

### 2.2 DIFFUSION MODELS

Diffusion models (Ho et al., 2020; Song et al., 2021a;b) construct a continuous time Markov process $X_t$ over a state space $\mathbb{X}$ with marginal distribution $p_t$ at time $t$ interpolating between a given target data distribution $p_0$ at time $t = 0$ and a reference noise distribution $p_1$ at time $t = 1$.

Let $\mathbb{P}_{t,t'}$ be the path-measure defining the law of paths $X_{[t,t']} = (X_s)_{s \in [t,t]}$ on the time interval $[t, t']$. We can equivalently express $X_{[t,t']}$ as a forward process $\overrightarrow{X}_s$ evolving $p_t$ forward in time from $t$ to $t'$, or as a backward process $\overleftarrow{X}_s$ evolving $p_{t'}$ backward in time from $t'$ to $t$. Since $\overrightarrow{X}_s$ and $\overleftarrow{X}'_s$ are constructed to be *time-reversals* of each other, we have forward and backward path measures coincide with $\mathbb{P}_{t,t'}$ and hence the Radon-Nikodym derivative between them equals 1 for any path $x_{[t,t']}$. It follows from Equation (2), the marginal densities ratio between $p_t(x_t)$ and $p_{t'}(x_{t'})$ can be expressed in terms of $R^{\mathbb{P}}_{t,t'}$, the RNE for the diffusion model over $[t, t']$,

$$p_{t'}(x_{t'})/p_t(x_t) = R^{\mathbb{P}}_{t,t'}(x_{[t,t']})^{-1}. \quad (4)$$

We describe two classes of diffusion models used in the literature when $\mathbb{X} = \mathbb{R}^d$ and when $\mathbb{X}$ is finite.

**Gaussian diffusions**  Gaussian diffusion models (Song et al., 2021b; Albergo et al., 2023) construct Markov processes over $\mathbb{X} = \mathbb{R}^d$. We define the forward process $\overrightarrow{X}_t$ obtained by integrating the forward SDE with drift $f_t$ and diffusion coefficient $\sigma_t$ initialised at the data distribution $p_0$ running forward in time, terminating at Gaussian noise $X_1 \sim p_1$,

$$\overrightarrow{X}_0 \sim p_0, \quad \overrightarrow{X}_t \sim p_t, \quad \mathrm{d}\overrightarrow{X}_t = f_t(\overrightarrow{X}_t)\,\mathrm{d}t + \sigma_t\,\mathrm{d}\overrightarrow{W}_t. \tag{5}$$

Similarly, the backward process $\overleftarrow{X}_t$ is obtained by integrating the backward SDE terminating at Gaussian noise $X_1 \sim p_1$ backward in time:

$$\overleftarrow{X}_1 \sim p_1, \quad \overleftarrow{X}_t \sim p_t, \quad \mathrm{d}\overleftarrow{X}_t = g_t(\overleftarrow{X}_t)\,\mathrm{d}t + \sigma_t\,\mathrm{d}\overleftarrow{W}_t, \tag{6}$$

where $g_t = f_t - \sigma_t^2 \nabla \log p_t$, and $\nabla \log p_t$ is the score function learned by a neural network.

**Discrete diffusions**  When $\mathbb{X}$ is finite, discrete diffusion models (Campbell et al., 2022; Lou et al., 2023; Shi et al., 2024) define $p_t$ as a probability vector of size $|\mathbb{X}|$ representing the law of $X_t$. The process is defined by integrating the forward CTMC initialised at the data distribution $p_0$ with rate matrix $\Lambda_t \in \mathbb{R}^{|\mathbb{X}| \times |\mathbb{X}|}$, terminating at a fully masked distribution $p_1$:

$$\overrightarrow{X}_0 \sim p_0, \quad \overrightarrow{X}_t \sim p_t, \quad \partial_t p_t = \Lambda_t^\top p_t. \tag{7}$$

The reverse process is encoded by the backward equation terminating at the fully masked distribution:

$$\overleftarrow{X}_1 \sim p_1, \quad \overleftarrow{X}_t \sim p_t, \quad \partial_t p_t = -\Lambda_t'^\top p_t. \tag{8}$$

Here, $\Lambda_t'$ is the backward rate matrix defined as $\Lambda_t'(x, y) = \Lambda_t(y, x)\frac{p_t(y)}{p_t(x)}$ for $y \neq x$ and $\Lambda_t'(x, x) = -\sum_{y \neq x} \Lambda_t'(x, y)$, where $\frac{p_t(y)}{p_t(x)}$ is known as the concrete score and is learned using a neural network.

## 2.3  REPLICA EXCHANGE AND ACCELERATED PT (APT)

Replica exchange, also known as parallel tempering (PT), was originally developed for sampling from multimodal distributions $\pi_0$ over $\mathbb{X}$. To do this, we first introduce an annealing path of distributions $(\pi_t)_{t \in [0,1]}$ over $\mathbb{X}$, interpolating between the target $\pi_0$ and reference $\pi_1$ chosen to be easy to sample from. PT construct a Markov chain $\mathbf{X}_n = (X_n^0, \cdots, X_n^M)$ over $\mathbb{X}^{M+1}$ invariant with respect to the product $\pi_{t_0} \times \cdots \times \pi_{t_M}$ where $0 = t_0 < \cdots < t_M = 1$ is an annealing schedule discretising the annealing path. Given $\mathbf{X}_n$ at time $n$ we generate $\mathbf{X}_{n+1}$ using a communication step and a local exploration step. The communication step performs sequence of Metropolised swaps moves between adjacent component of $\mathbf{X}_n$. It is advantageous to apply a *non-reversible* communication (Okabe et al., 2001; Syed et al., 2022): the swap between components $m - 1$ and $m$ of $\mathbf{X}_n$ is proposed only at iterations $n$ with matching parity, $m \equiv n \mod 2$. Then, the local exploration step updates each $m$-th component with a MCMC move targeting $\pi_{t_m}$. Both local exploration and communication can be carried out in parallel for each $m$.

The standard formulation of PT proposes swapping neighbouring samples directly. This becomes inefficient when the neighbouring distributions of the annealing sequence have little overlap. To address this issue, Ballard & Jarzynski (2009; 2012); Zhang et al. (2025) proposed APT, extending the communication step to the *path space* of stochastic processes. Concretely, an accelerated PT (APT) swap move between states $(x, x')$ targeting $\pi_t$ and $\pi_{t'}$ respectively simulates (1) a *forward proposal* Markov process $\overrightarrow{X}_s$, and (2) a *backward proposal* Markov process $\overleftarrow{X}'_s$ over some time interval $s \in [t, t']$. The forward proposal $\overrightarrow{X}_s$ propagates $\overrightarrow{X}_t = x$ forward in time from $t$ to $t'$, and the backward proposal propagates $\overleftarrow{X}'_{t'} = x'$ backward in time from $t'$ to $t$. We then replace $(x, x')$ with the terminate states $(X'_t, X_{t'})$ with probability $\alpha_{t,t'}(\overrightarrow{X}_{[t,t']}, \overleftarrow{X}'_{[t,t']})$ equal to,

$$\alpha_{t,t'}(x_{[t,t']}, x'_{[t,t']}) = \min\left\{1, \frac{\mathrm{d}\overleftarrow{\mathbb{Q}}'_{t,t'}}{\mathrm{d}\overrightarrow{\mathbb{Q}}_{t,t'}}(x_{[t,t']})\frac{\mathrm{d}\overrightarrow{\mathbb{Q}}_{t,t'}}{\mathrm{d}\overleftarrow{\mathbb{Q}}'_{t,t'}}(x'_{[t,t']})\right\}. \tag{9}$$

Here $\overrightarrow{\mathbb{Q}}_{t,t'}$ denotes the law of the forward paths $\overrightarrow{X}_{[t,t']} = (\overrightarrow{X}_s)_{s \in [t,t']}$ initialised at $\overrightarrow{X}_t \sim \pi_t$ and $\overleftarrow{\mathbb{Q}}'_{t,t'}$ denotes the law of the backward paths $\overleftarrow{X}'_{[t,t']} = (\overleftarrow{X}'_s)_{s \in [t,t']}$ terminating at $\overleftarrow{X}'_{t'} \sim \pi_{t'}$. The swap move remains valid when the paths are discretised as long as the simulation of the proposal and the calculation of the Radon-Nikodym derivative follow the same discretisation.

---

**Algorithm 1** CREPE: Control with REPlica Exchange

---

**Inputs:** $J$ pretrained diffusions; annealing path $(\pi_t)_{t \in [0,1]}$, discretisation schedule $(t_m)_{m=0}^M$; PT iterations $N$.
**Output:** target samples $\{\mathbf{X}_n\}_{n=1}^N$.

> Initialise $\mathbf{X}_0 = (X_0^0, \ldots, X_0^M)$ by running diffusion model.
> **for** $n = 1, \ldots, N$ **do**        ▷ *Run PT.*
>> $\mathbf{X}_n = (X_n^0, \ldots, X_n^M) \leftarrow \mathbf{X}_{n-1}$
>> **for** $m \equiv n \bmod 2$ **do**        ▷ *Communication (parallelise)*
>>> $t \leftarrow t_{m-1}$ and $t' \leftarrow t_m$        ▷ *Diffusion time interval*
>>> $\overrightarrow{X}_t \leftarrow X_n^{m-1}$ and $\overleftarrow{X}'_{t'} \leftarrow X_n^m$        ▷ *Generate proposal paths*
>>> Simulate proposal paths $\overrightarrow{X}_s$ and $\overleftarrow{X}'_s$ for $s \in [t, t']$        ▷ *Equations (10) and (11)*
>>> Compute $R_{t,t'}^{\mathbb{Q}}, R_{t,t'}^{\mathbb{P}^1}, \ldots, R_{t,t'}^{\mathbb{P}^J}$ for $\overrightarrow{X}_{[t,t']}$ and $\overleftarrow{X}'_{[t,t']}$        ▷ *See Appendix A.2*
>>> $\alpha_{t,t'} \leftarrow \alpha_{t,t'}(\overrightarrow{X}_{[t,t']}, \overleftarrow{X}'_{[t,t']})$        ▷ *See Equation (13)*
>>> $(X_n^{m-1}, X_n^m) \leftarrow (\overleftarrow{X}'_t, \overrightarrow{X}_{t'})$ with probability $\alpha_{t,t'}$.        ▷ *Swap move*
>> (Optionally) Update $X_n^m, m = 0, \ldots, M$ with local exploration.        ▷ *Local Exploration (parallelise)*

---

# 3 METHODS

Given diffusion models $p_t^j$ for the data distributions $p_0^j$ for $j = 1, \ldots, J$, we aim to generate samples from a new distribution $\pi_0$ related to $p_0^j$ without retraining the model. Some common examples of such tasks include tempering, reward-tilting/posterior sampling, and model composition.

| | |
|---|---|
| **tempering:** | $\pi_0(x) \propto p_0^j(x)^\beta$ with inverse-temperature $\beta > 0$; |
| **reward-tilting/posterior sampling:** | $\pi_0(x) \propto p_0^j(x) \exp(r_0(x))$ with reward/likelihood $r_0(x)$; |
| **model composition:** | $\pi_0(x) \propto \prod_j p_0^j(x)$ composing $J$ diffusions $p_0^j, j = 1, \cdots, J$. |

These are not the only options. In fact, one can also combine these tasks. For example, debiased **classifier-free guidance** aiming to sample from $\pi_0(x) \propto p_0(x)^{1-w} p_0(x \mid c)^w$, can be achieved by combining tempering with composition . We will also demonstrate other combinations in Section 4.

This section shows how this can be achieved using the accelerated parallel tempering framework outlined in Section 2.3. We can adapt this to obtain the Control REPlica Exchange (CREPE) algorithm summarised in Algorithm 1. We will outline the ingredients for CREPE, i.e. (1) an annealing path, (2) a communication move, and (3) a local exploration move.

## 3.1 ANNEALING PATH, COMMUNICATION AND LOCAL EXPLORATION

**Annealing path** We first introduce an annealing path of distributions $(\pi_t)_{t \in [0,1]}$ interpolating between the target distribution $\pi_0$ when $t = 0$ and a reference distribution where inference is tractable $\pi_1$ when $t = 1$. For example, we can assume $\pi_1$ is a Gaussian in the case of Gaussian diffusion or a fully masked distribution in the discrete diffusion case. We additionally assume we can express the marginal density ratio of the annealing path $\pi_t$ as functions of the marginal density ratio for the pre-trained diffusion model $p_t^j$ so that one can plug in the RNE relation in Equation (4). Some common examples of annealing path include for inference time control include:

| | |
|---|---|
| **tempering:** | $\pi_t(x) \propto p_t^j(x)^\beta$ with inverse-temperature $\beta > 0$; |
| **reward-tilting/posterior sampling:** | $\pi_t(x) \propto p_t^j(x) \exp(r_t(x))$ with reward/likelihood $r_t(x)$; |
| **model composition:** | $\pi_t(x) \propto \prod_j p_t^j(x)$ composing $J$ diffusions $p_t^j, j = 1, \cdots, J$. |

**Communication** We now introduce the forward and backward proposal processes in the APT framework and show how to compute the acceptance probability. Here we focus on our discussion on the communication between two distributions $\pi_t$ and $\pi_{t'}$, and the same formula applies to any pair.

We first introduce the proposal processes for the communication. Precisely, we introduce Markov processes $\overrightarrow{X}_s$ and $\overleftarrow{X}'_s$, so that we can simulate the dynamics forward and backward in time, respectively. Concretely, in the case of Gaussian diffusions defined in Equations (5) and (6), we introduce the forward and backward SDEs driven by the same noise $\sigma_t$ but with user-specified drift $a_t$, and $b_t$. In the case of a discrete diffusion, we introduce a forward and backward CTMC with user-specified

rate matrices $A_t$ and $B_t$,

$$\text{Forward proposal:} \quad \mathrm{d}\overrightarrow{X}_t = a_t(\overrightarrow{X}_t)\mathrm{d}t + \sigma_t \mathrm{d}\overrightarrow{W}_t, \qquad \partial_t q_t = A_t^\top q_t \tag{10}$$

$$\text{Backward proposal:} \quad \mathrm{d}\overleftarrow{X}'_t = b_t(\overleftarrow{X}'_t)\mathrm{d}t + \sigma_t \mathrm{d}\overleftarrow{W}_t, \qquad \partial_t q'_t = -B_t^\top q'_t \tag{11}$$

Here we use the arrow to indicate that the forward and backward processes correspond to different random variables, each with its own marginal density, and they are not necessarily time-reversal of each other. In fact, there is considerable flexibility in choosing $a_t$ and $b_t$ (or, in the discrete case, $A_t$ and $B_t$). A natural choice for the inference-time control task is to modify the diffusion dynamics so that $\overrightarrow{X}_t$ and $\overleftarrow{X}_t$ approximate $\pi_t$ at time $t$ provided when $\overrightarrow{X}_0 \sim \pi_0$ and $\overleftarrow{X}'_1 \sim \pi_1$. However, in many cases, this choice is intractable or doesn't exist. We instead apply an approximation as exemplified in Appendix A.3. The misalignment with $\pi_t$ is corrected through the acceptance probability.

We now focus on the proposal processes within the time interval $[t, t']$. We consider we have samples at $\pi_t$ and $\pi_{t'}$ and perform communication steps between them using the APT framework. We define $\overrightarrow{\mathbb{Q}}_{t,t'}$ as the law of the path $\overrightarrow{X}_{[t,t']}$ obtained by evolving samples from $\pi_t$ at time $t$ to time $t'$ using the forward proposal process. Similarly we define $\overleftarrow{\mathbb{Q}}'_{t,t'}$ as the law of the path $\overleftarrow{X}'_{[t,t']}$ obtained by evolving samples from $\pi_{t'}$ at time $t'$ backward to time $t$ using the backward proposal process. We highlight again that we do not require $\overrightarrow{X}_{[t,t']}$ is a time reversal of $\overleftarrow{X}'_{[t,t']}$; we also do not assume the simulated forward and backward trajectories terminate at $\pi_{t'}$ or $\pi_t$ respectively.

We can use Equation (2) to express the Radon-Nikodym derivative between $\overleftarrow{\mathbb{Q}}'_{t,t'}$ and $\overrightarrow{\mathbb{Q}}_{t,t}$ in terms of the marginal density ratio of $\pi_{t'}(x_{t'})$ and $\pi_t(x_t)$ and $R^{\mathbb{Q}}_{t,t'}(x_{[t,t']})$, the RNE for the proposal processes over $[t, t']$, for any path $x_{[t,t']}$ generated by the proposals,

$$\frac{\mathrm{d}\overleftarrow{\mathbb{Q}}'_{t,t'}}{\mathrm{d}\overrightarrow{\mathbb{Q}}_{t,t'}}(x_{[t,t']}) = \frac{\pi_{t'}(x_{t'})}{\pi_t(x_t)} R^{\mathbb{Q}}_{t,t'}(x_{[t,t']}). \tag{12}$$

We can substitute Equation (12) into Equation (9) to obtain the acceptance probability,

$$\alpha_{t,t'}(x_{[t,t']}, x'_{[t,t']}) = \min\left\{1, \frac{\pi_{t'}(x_{t'})}{\pi_t(x_t)} \cdot \frac{\pi_t(x'_t)}{\pi_{t'}(x'_{t'})} \cdot \frac{R^{\mathbb{Q}}_{t,t'}(x_{[t,t']})}{R^{\mathbb{Q}}_{t,t'}(x'_{[t,t']})}\right\}. \tag{13}$$

Provided we can express the marginal density ratio of the $\pi$ in terms of the marginal density ratio of the pretrained diffusion models $p^1, \ldots, p^j$, we can compute Equation (13) in terms of the RNE's for the pre-trained diffusion model, $R^{\mathbb{P}^j}_{t,t'}, \ldots, R^{\mathbb{P}^j}_{t,t'}$, using the RNE relation Equation (4). For example in the case of tempering with $\pi_t(x) \propto p^j_t(x)^\beta$, we have $\pi_{t'}(x_{t'})/\pi_t(x_t) \propto R^{\mathbb{P}^j}_{t,t'}(x_{[t,t']})^{-\beta}$. See Appendix A.5 for explicit expressions of the acceptance probability for tempering, reward-tilting/posterior sampling, and model composition. We can tractably compute the RNE terms in Equation (13) via the transition-kernel product in Equation (3) or via the path-integral in Equations (15) and (19) in Appendix A.2.

**Local exploration**  Optionally, we can apply local exploration after each communication step[1]. For Gaussian diffusion, we adopt the *corrector* step from the predictor–corrector algorithm of Song et al. (2021b), using the score function of $\pi_t$ instead of $p_t$.

Additionally, for the highest time step $t = 1$ in Gaussian diffusions, the target marginal $\pi_1$ is a Gaussian distribution. To accelerate mixing, we resample directly from this Gaussian instead of performing a Langevin step following Syed et al. (2021; 2022); Zhang et al. (2025).

For CTMC, the concrete score defines the density ratio between two states, allowing for the direct application of the Metropolis–Hastings algorithm (Metropolis et al., 1953; Hastings, 1970). We include a detailed discussion on the local move in Appendix A.6.

---

[1]Note that in standard parallel tempering, this local exploration is essential because its communication step only involves a deterministic proposal. In contrast, accelerated PT, and thus our framework, uses a stochastic communication proposal, which already introduces randomness. The local move only provides additional refinement and hence can be omitted when the scores of $\pi_t$ are prohibitively expensive.

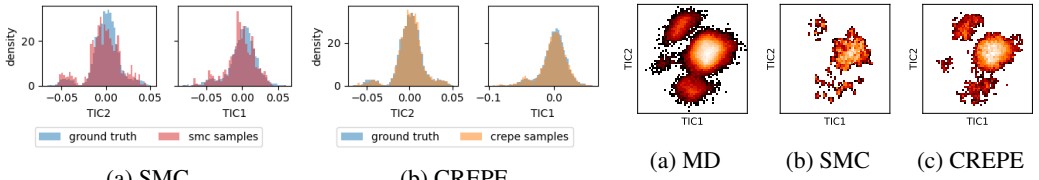

(a) SMC          (b) CREPE        (a) MD    (b) SMC    (c) CREPE

Figure 3: Histogram of Alanine Hexapeptide annealed to 600K by SMC and CREPE. 600K by SMC and CREPE, projected to two TICA axes. CREPE maintains more diversity.

Figure 4: TICA of Alanine Hexapeptide annealed to

## 3.2 CONTROL WITH REPLICA EXCHANGE (CREPE)

Now, we put the ingredients together in Algorithm 1. Letting the schedule of times $0 = t_0 < \cdots < t_M = 1$, we generate a Markov chain $\mathbf{X}_n = (X_n^0, \ldots, X_n^m)$ in $\mathbb{X}^{M+1}$ targeting $\pi_{t_0} \times \cdots \times \pi_{t_M}$ using the accelerated PT algorithm described in Section 2, with the annealing path, communication step, and local exploration move described above.

In practice, we can stabilise PT by running it only up to a small time step $t_0 > 0$, instead of across the entire diffusion process. After $t_0$, we will directly continue sampling until 0 with the diffusion model using drift $a_t$. This is because tiny time steps often introduce numerical instability and yield low acceptance rates, especially in high-dimensional spaces. By truncating PT early, we avoid these issues while retaining effectiveness, as the denoising steps, after sufficiently small $t_0$, will have no semantic change to the sample. This strategy was also applied in inference-time control with SMC, where resampling is restricted to a limited time interval (Skreta et al., 2025).

## 3.3 RELATED WORKS

SMC has been extensively applied to steer the generation process (Wu et al., 2023; Dou & Song, 2024; Singhal et al., 2025; Skreta et al., 2025; Lee et al., 2025; Pani et al., 2025; He et al., 2025c; Hasan et al.). SMC based-methods simulate the discretised annealing path sequentially and generates particles in parallel. CREPE introduces a related, but computationally dual perspective on inference time control, by simulating the discretised annealing path in parallel and generating particles sequentially via MCMC. We illustrate their difference and connection in Figure 2.

**Trade-offs between SMC and CREPE** SMC typically requires a large batch of particles to run in parallel, can suffer from low sample diversity or even mode collapse when the batch size is small, (Li et al., 2024; Young & Akyildiz, 2024; Lee et al., 2025). A common strategy is to run multiple batches. While this can improve diversity, it does not address the bias introduced by a small batch size. We illustrate and discuss this in detail in Appendix A.1. CREPE, by contrast, only requires parallelisation across different diffusion times $(t_m)_{m=0}^{M}$, but generates new samples sequentially. It tends to produce more diverse samples naturally, as we demonstrate in experiments. Another advantage of CREPE compared to SMC-based methods is that it supports online refinement: new constraints can be introduced on the fly, or samples can be further refined if their quality is insufficient. It is also an anytime inference algorithm: unlike SMC, which returns target samples only after the final iteration, CREPE can terminate at any iteration. However, *a burn-in period is required*: the samples from the first several iterations may not follow the desired target $\pi_0$ and may be discarded.

We also highlight that when the number of PT iterations equals the number of SMC particles, and both methods use the same discretisation steps, controlling with PT and SMC incur the same number of network function evaluations (NFEs). We provide a detailed discussion in Appendix A.7.

## 4 EXPERIMENTS

We evaluate our proposed algorithm through comprehensive experiments across various domains, including molecules, images, trajectories, and discrete data. Please refer to Appendix B for details.

**Inference-time Tempering for Boltzmann Sampling** We first consider the tempering task for sampling from Boltzmann distributions. Concretely, assuming we have a pretrained diffusion model trained on samples from $p_0(x) \propto \exp(-U(x)/k_B T_{\text{high}})$, we aim to generate samples from $\pi_0(x) \propto \exp(-U(x)/k_B T_{\text{low}})$. This setting was considered in Skreta et al. (2025); Rissanen et al.

Table 1: Inference-time tempering performance for Alanine Dipeptide, Tetrapeptide and Hexapeptide.

| | | | FKC | | RNE | CREPE |
|---|---|---|---|---|---|---|
| | | | Anneal Score | Anneal Noise | | (Ours) |
| Alanine Dipeptide | **ALA Dipeptide** (800K → 300K) | Energy TVD | $0.345_{\pm 0.010}$ | $0.894_{\pm 0.002}$ | $0.391_{\pm 0.006}$ | $\textbf{0.224}_{\pm 0.005}$ |
| | | Distance TVD | $0.023_{\pm 0.001}$ | $0.036_{\pm 0.001}$ | $0.024_{\pm 0.001}$ | $\textbf{0.019}_{\pm 0.000}$ |
| | | Sample W2 | $0.293_{\pm 0.001}$ | $0.282_{\pm 0.001}$ | $0.282_{\pm 0.001}$ | $\textbf{0.264}_{\pm 0.001}$ |
| | | TICA MMD | $0.116_{\pm 0.003}$ | $0.108_{\pm 0.004}$ | $0.168_{\pm 0.007}$ | $\textbf{0.096}_{\pm 0.014}$ |
| Alanine Tetrapeptide | **ALA Tetrapeptide** (800K → 500K) | Energy TVD | $\textbf{0.122}_{\pm 0.012}$ | $0.436_{\pm 0.007}$ | $0.154_{\pm 0.006}$ | $\textbf{0.122}_{\pm 0.004}$ |
| | | Distance TVD | $\textbf{0.014}_{\pm 0.000}$ | $0.015_{\pm 0.000}$ | $\textbf{0.013}_{\pm 0.001}$ | $\textbf{0.013}_{\pm 0.001}$ |
| | | Sample W2 | $0.923_{\pm 0.008}$ | $0.892_{\pm 0.001}$ | $0.893_{\pm 0.005}$ | $\textbf{0.856}_{\pm 0.004}$ |
| | | TICA MMD | $0.183_{\pm 0.020}$ | $0.138_{\pm 0.017}$ | $0.155_{\pm 0.009}$ | $\textbf{0.035}_{\pm 0.002}$ |
| Alanine Hexapeptide | **ALA Hexapeptide** (800K → 600K) | Energy TVD | $0.091_{\pm 0.006}$ | $0.206_{\pm 0.005}$ | $\textbf{0.087}_{\pm 0.003}$ | $0.398_{\pm 0.001}$ |
| | | Distance TVD | $0.018_{\pm 0.000}$ | $0.020_{\pm 0.001}$ | $\textbf{0.010}_{\pm 0.001}$ | $\textbf{0.009}_{\pm 0.001}$ |
| | | Sample W2 | $1.585_{\pm 0.001}$ | $1.652_{\pm 0.012}$ | $1.618_{\pm 0.001}$ | $\textbf{1.299}_{\pm 0.004}$ |
| | | TICA MMD | $0.088_{\pm 0.004}$ | $0.068_{\pm 0.010}$ | $0.042_{\pm 0.004}$ | $\textbf{0.009}_{\pm 0.001}$ |

Table 2: Debias ImageNet-64 CFG. We do not discard burn-in samples in CREPE to ensure a fair comparison.

| Method | #Samples | IR (↑) | CLIP (↑) | FID (↓) |
|---|---|---|---|---|
| FKC | 8 | -0.29 | 24.17 | 1.85 |
| | 32 | -0.14 | 23.98 | 1.84 |
| | 128 | -0.03 | 24.04 | 1.89 |
| | 512 | -0.08 | 24.31 | 1.96 |
| CREPE | 8 | -0.30 | 24.10 | 1.92 |
| | 32 | -0.21 | 24.21 | 1.88 |
| | 128 | -0.09 | **24.37** | 1.86 |
| | 512 | **0.09** | 24.28 | **1.79** |

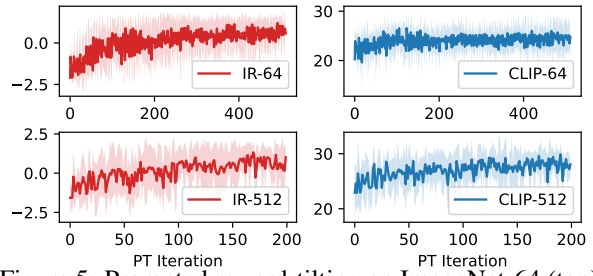

Figure 5: Prompted reward-tilting on ImageNet-64 (top) and 512 (bottom). We report mean and std across five different classes and prompts.

(2025); Akhound-Sadegh et al. (2025) to accelerate Boltzmann sampling tasks. In Table 1, we evaluate our proposed algorithm on three molecules with different sizes: Alanine Dipeptide, Tetrapeptide and Hexapeptide. We report the total variant distances (TVD) of energy and distance histograms, W2 distance of samples, as well as the maximum mean discrepancy (MMD) for the sample projected to 2D space with time-lagged independent component analysis (TICA, Molgedey & Schuster, 1994). We also show the TICA plot for Hexapeptide samples generated by SMC (w. RNE) and CREPE, along with molecular dynamics (MD) samples in Figures 3 and 4. We can see CREPE achieves superior performance on three of the targets across most metrics. In particular, Figure 3 shows that CREPE achieves lower bias compared to SMC, echoing our discussion on mini-batch SMC in Appendix A.1. Figure 4 shows that CREPE maintains better diversity and avoids missed modes. The only exception is the energy of the alanine hexapeptide. A likely explanation is that the pretrained model incurs higher error on this molecule, which is amplified by PT through repeated iterations. We also note that the energy histogram alone may not always reflect overall performance. It may appear favourable even when mode collapsing, as observed by Blessing et al. (2024); He et al. (2025a).

**Debiasing Classifier-Free Guidance for Image Generation**  We now consider applying CREPE to debias classifier-free guidance (CFG, Ho & Salimans, 2022), a setting also explored by Skreta et al. (2025) with SMC. Concretely, given an unconditional diffusion ($p_t$) and an conditional diffusion ($p_0(\cdot|c)$), we aim to sample from $\pi_0(\cdot) \propto p_0(\cdot)^{1-w}p_0(\cdot|c)^w$. In Table 2, we evaluate the ImageReward (IR, Xu et al., 2023), CLIP score (Hessel et al., 2021) and FID (Heusel et al., 2017) for images generated by CREPE. The IR and CLIP are conditioning on the class label. We also report results by FKC (Skreta et al., 2025), which is based on SMC for debiasing the CFG. We can see that when the number of samples is small (e.g., 8), the SMC-based FKC outperforms CREPE as expected, since PT typically requires a burn-in period. However, as the number of generated samples increases, CREPE empirically outperforms FKC, particularly in terms of FID. Additionally, in the example images shown in Figures 11 to 14, we can see that FKC tends to produce visually similar samples within a batch, whereas CREPE maintains higher diversity.

**Reward-tilting for Image Generation**  We now turn to reward-tilting in the context of image generation. Specifically, we generate class-conditioned images using CFG with debiasing, and further steer the generation with more detailed instructions provided by ImageReward through a prompt. This also serves as an example of combining multiple tasks (debiasing CFG and reward-tilting).

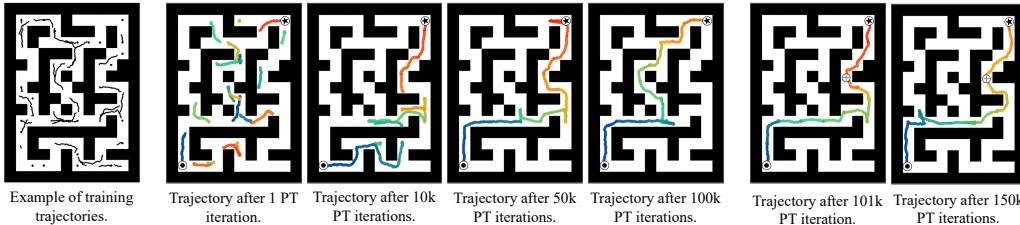

| Example of training trajectories. | Trajectory after 1 PT iteration. | Trajectory after 10k PT iterations. | Trajectory after 50k PT iterations. | Trajectory after 100k PT iterations. | Trajectory after 101k PT iteration. | Trajectory after 150k PT iterations. |

Figure 6: Stitched trajectory by CREPE with online refinement. We also visualise the training dataset (leftmost plot). In the first 100k iterations, the trajectories navigate from the initial ⊙ to the final target ✷. Starting from 100k iteration, an additional intermediate point ⊕ is introduced. We observe that this new constraint is quickly satisfied (only 1k iterations after the new reward is added).

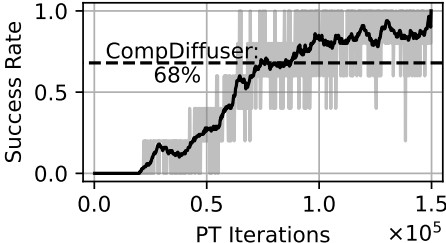
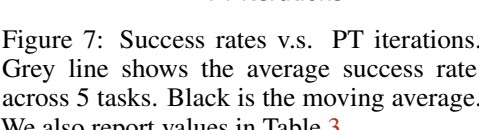

Figure 7: Success rates v.s. PT iterations. Grey line shows the average success rate across 5 tasks. Black is the moving average. We also report values in Table 3.

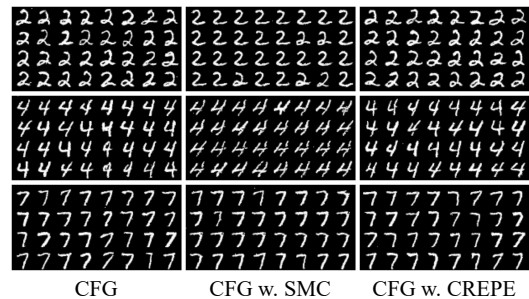

Figure 8: MNIST samples generated by CFG, and debiased by SMC and CREPE.

We visualise the samples obtained along the PT chains (thinned every 8 iterations) with their class label and prompt in Figure 1. We also plot IR and CLIP scores along PT iterations across five different classes and prompts in Figure 5. The IR and CLIP are conditional on the prompt. After the first burn-in period, CREPE effectively produces diverse images that closely align with the prompt.

**Model Composition with Reward for Maze Navigation**  We now consider model composition. Following Luo et al. (2025), we compose diffusion models trained on short trajectories to synthesise a coherent long-horizon path through the maze. Unlike Luo et al. (2025), who train diffusion models conditioned on both ends and stitch segments via conditioning, we train an unconditional model and stitch segments using a reward function. This task can be viewed as a combination of reward-tilting and model composition, where we aim to generate samples from $\pi_0([x^{(1)}, x^{(2)}, \cdots, x^{(J)}]) = \exp(r) \prod_j p_0^j(x^{(j)})$. This reward-based composition affords flexible constraints on the trajectory.

We use the `pointmaze-giant-stitch-v0` dataset from Park et al. (2024), which consists of short trajectories of length 64 in a large 2D maze. We train an unconditional diffusion model on these short trajectories. To generate stitched trajectories, we impose the following rewards: (1) the starting point of the first trajectory is sufficiently close to the initial state, (2) the endpoint of the last trajectory is sufficiently close to the target state, and (3) consecutive trajectories are connected in a tail-to-head manner. We include detailed forms of the reward in Appendix B.4. We first evaluate our approach on the five different initial–goal pairs considered by Luo et al. (2025). We report the success rate in Table 3 and visualise the corresponding trace plots in Figure 7. We include the results by (Luo et al., 2025) as a reference. As we can see, combining an unconditional diffusion model with CREPE achieves comparable or even better performance than directly training a conditional model, at the cost of more computing resources.

**CREPE with Online Refinement**  An advantage of training an unconditional model and stitching with CREPE is that it offers greater flexibility in specifying the reward function, and it naturally extends to online settings where new constraints may be introduced on the fly. To show this, we first run CREPE to navigate from the initial to the final target, and then add an additional reward that requires the trajectory to pass through an intermediate point. In Figure 6, we visualise stitched trajectory samples produced by CREPE at different PT iterations, where the intermediate point is introduced at 100k iterations. We can see the new constraint is quickly satisfied.

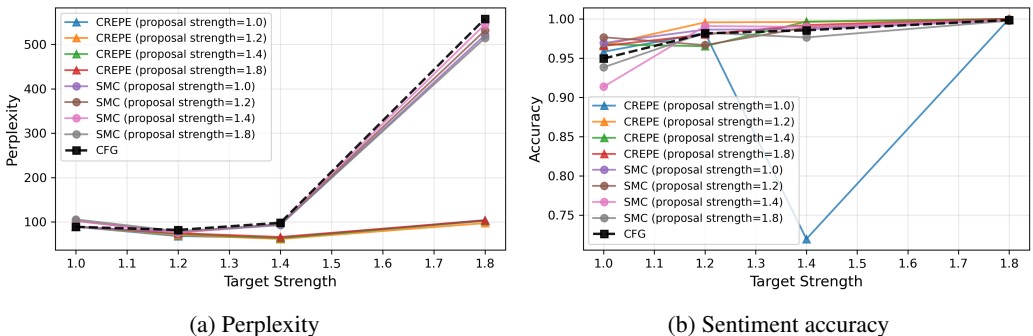

(a) Perplexity

(b) Sentiment accuracy

Figure 9: Debiasing CFG for CTMC on text with SMC and CREPE. Perplexity (left) and sentiment accuracy (right) as a function of target strength for SMC and CREPE across different proposal strengths. As we can see, CREPE achieves a significant reduction in perplexity while maintaining high sentiment accuracy across most settings.

**CREPE on CTMC** We now consider applying CREPE to discrete diffusion. In the first example, we consider debiasing CFG for mask diffusion, as considered by Lee et al. (2025). In Figure 8, we visualise samples obtained by SMC (Lee et al., 2025) and CREPE. We can see that both algorithm achieves more plausible samples, with CREPE presenting slightly more sample diversity.

In the second example, we consider debiasing the CFG for sentiment-controlled text generation using discrete diffusion. The goal is to generate text with a specified sentiment (positive or negative) by conditioning on the prompt "The sentiment of the text is {negative|positive}". We sweep over target CFG strengths $\in \{1.0, 1.2, 1.4, 1.8\}$. Also, since our framework allows arbitrary proposal processes, we sweep over different proposal CFG strengths $\in \{1.0, 1.2, 1.4, 1.8\}$.

Figure 9 compares SMC and CREPE across the target strength sweep. CREPE maintains substantially better perplexity throughout the range, improving by up to a factor of 5 at target strength $=$ 1.8. Both methods achieve similar sentiment accuracy, except for CREPE at (target strength $=$ 1.4, proposal strength $=1.0$) where accuracy drops. A possible explanation is that CREPE iteratively refines the current samples, so a poor sample can destabilize performance, whereas mini-batch SMC runs independent batches and is more robust to occasional failures. This suggests that CREPE is not immune to instabilities, and further investigation is needed, especially for its application to text. To better understand the gain from CREPE, in Appendix D, we visualize the running average of perplexity and accuracy over CREPE iterations. We observe that perplexity steadily decreases. This indicates that a standard CFG can distort the text distribution, and CREPE effectively debiases the CFG-induced distortion, producing higher-quality samples with substantially lower perplexity.

## 5 CONCLUSIONS AND FUTURE WORKS

In this work, we propose CREPE, a new framework for controlling diffusion models using replica exchange. CREPE offers an alternative to the widely employed SMC-based approaches for a broad range of inference-time control tasks for diffusion models across different modalities. It demonstrates comparable efficiency with SMC, while additionally supporting online refinement and maintaining better sample diversity, opening a new avenue for further exploration. One potential direction is to investigate how advanced schedule-tuning and path–selection techniques (Masrani et al., 2021; Syed et al., 2021; Surjanovic et al., 2022; Máté & Fleuret, 2023; York, 2023) from classical parallel tempering for sampling can be adapted to our setting to provide better control over diffusion models.

The main limitations of CREPE are the presence of a burn-in period and approximation errors introduced in the communication acceptance rate: CREPE typically requires a burn-in period to reach optimal performance. For large systems and expensive networks, this can result in a high computational cost. Additionally, CREPE relies on the assumption of a perfect diffusion model without discretisation error, which often does not hold in practice. While we did not observe major failures in our experiments, this assumption may break in other settings. Besides, these approximation errors can accumulate over iterations, leading to deviations from the desired target.

## LLM USAGE DISCLOSURE

LLM was used at the sentence level to correct grammar.

## ACKNOWLEDGEMENTS

JH acknowledges support from the University of Cambridge Harding Distinguished Postgraduate Scholars Programme. PP and LZ are supported by the EPSRC CDT in Modern Statistics and Statistical Machine Learning (EP/S023151/1). JMHL acknowledge support from EPSRC funding under grant EP/Y028805/1. JMHL also acknowledges support from a Turing AI Fellowship under grant EP/V023756/1. SS acknowledges support from the NSERC Postdoctoral Fellowship. On a different note, JH and FV gratefully acknowledge PJ for the excellent food recommendations.

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

# APPENDIX

# A  SUPPLEMENTARY METHODS AND DISCUSSION

## A.1  WHY MINI-BATCH SMC IS NOT FAVOURABLE?

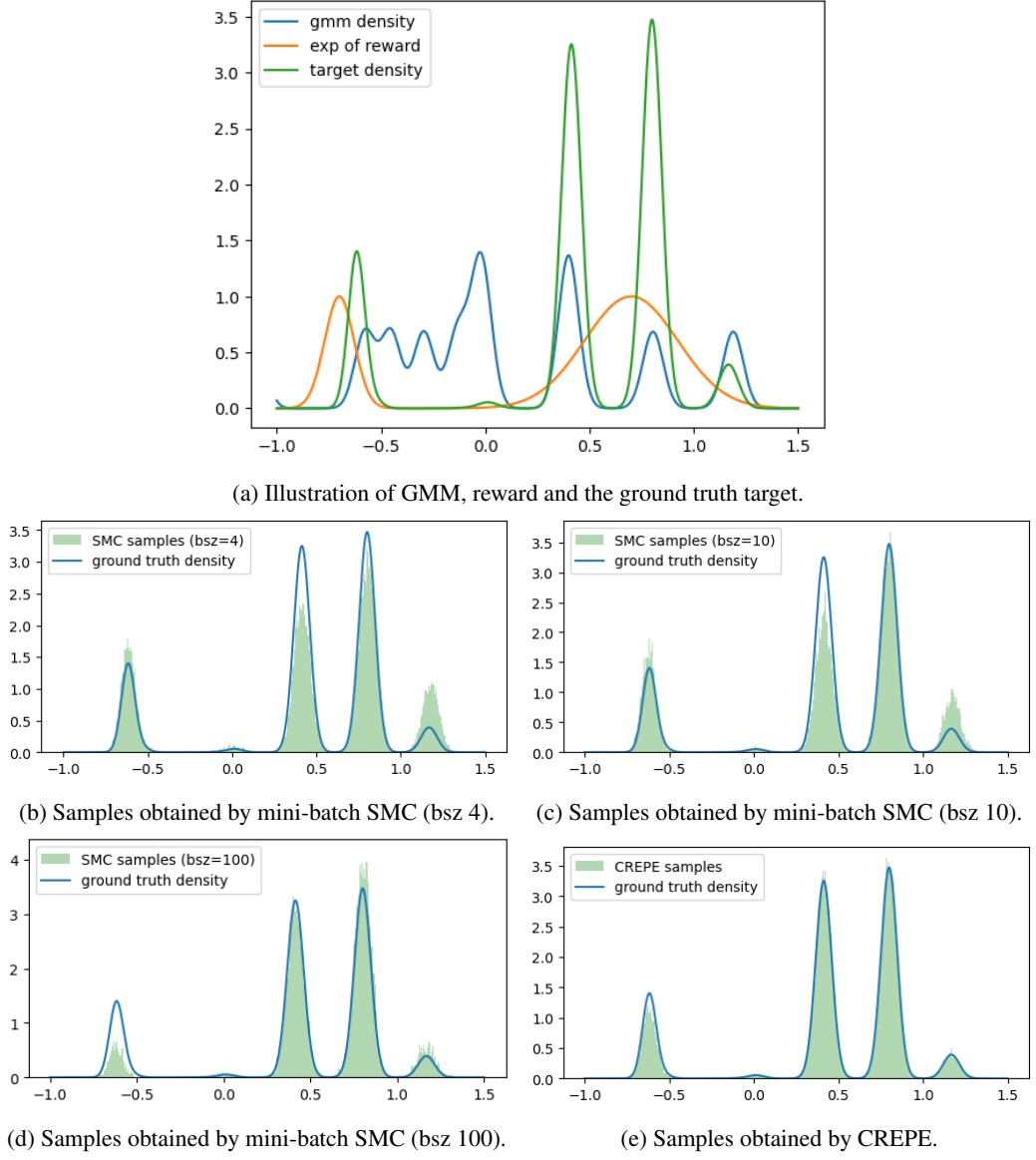

(a) Illustration of GMM, reward and the ground truth target.

(b) Samples obtained by mini-batch SMC (bsz 4).    (c) Samples obtained by mini-batch SMC (bsz 10).

(d) Samples obtained by mini-batch SMC (bsz 100).    (e) Samples obtained by CREPE.

Figure 10: Illustration of reward-tilted samples obtained with mini-batch SMC and CREPE. For SMC, we collect 150,000 samples with a batch size of 4/10/100; while for CREPE, we collect 150,000 samples in a single run.

In our main experiments, we observe that CREPE produces samples with higher diversity than SMC. A natural question is: *why not simply run many mini-batches of SMC to recover diversity?*

In fact, our SMC baseline already uses this strategy: following the convention in the diffusion-control-with-SMC literature (e.g. Skreta et al., 2025), we run SMC in mini-batches and aggregate the resulting samples. Even in this setting, SMC still underperforms CREPE in terms of diversity. One might further ask why we do not reduce the batch size even more and increase the number of SMC runs.

The reason is that this comes at the cost of a large bias. Running many mini-batch SMCs is conceptually equivalent to repeatedly restarting a local sampling algorithm. To see why this is problematic, consider a multimodal target distribution where each SMC run tends to collapse onto a single mode due to resampling. If we restart SMC many times, we may indeed obtain samples from different modes, but the relative mode weights now reflect the probability that a single SMC run collapses into each mode, rather than the true target mass of each mode. Thus, while the aggregated samples may appear diverse, the relative weights between modes are incorrect, and the resulting empirical distribution is systematically biased away from the true target.

To illustrate this bias, we visualise reward-tilting on a 1D Gaussian mixture in Figure 10. As shown in the figure, while mini-batch SMC can produce samples from all modes, the relative proportions of these modes are noticeably distorted, whereas CREPE preserves the true mode weights much more faithfully. We note that this is only a 1D toy example; in higher dimensions, samples from SMC in each batch will collapse further, and the misweighting becomes even more severe.

## A.2 CONTINUOUS FORM FOR $R$

We can express the RNE defined in Equation (3) analytically for SDE or CTMC (Berner et al., 2025; Holderrieth et al., 2025): Consider the following forward and backward SDEs:

$$\mathrm{d}\overrightarrow{X}_s = \mu_s(\overrightarrow{X}_s)\,\mathrm{d}s + \sigma_s\,\mathrm{d}\overrightarrow{W}_s, \quad \mathrm{d}\overleftarrow{X}'_s = \nu_t(\overleftarrow{X}'_s)\,\mathrm{d}s + \sigma_s\,\mathrm{d}\overleftarrow{W}_s \tag{14}$$

Then for any valid trajectory $X_{[t,t']}$ within time-interval $[t, t']$, under mild condition, we have

$$R_{t,t'}(x_{[t,t']}) = \exp\left(\int_t^{t'} \frac{1}{\sigma_t^2}\nu_s(x_s)\cdot\mathrm{d}\overleftarrow{x_s} - \int_t^{t'} \frac{1}{\sigma_t^2}\mu_s(x_s)\cdot\mathrm{d}\overrightarrow{x_s} + \frac{1}{2}\int_t^{t'} \frac{1}{\sigma_s^2}\left(\|\mu_s(x_s)\|^2 - \|\nu_s(x_s)\|^2\right)\mathrm{d}s\right), \tag{15}$$

where the first and second terms on the RHS represent the Itô forward and backward integral. Consider discretisation with $t = t_0 < t_1 < \cdots < t_K = t'$, they are defined as:

$$\int_t^{t'} a_s(x_s)\cdot\mathrm{d}\overrightarrow{x_s} = \lim_{\max_k |t_{k+1}-t_k|\to 0}\sum_k a_{t_k}(x_{t_k})\cdot(x_{t_{k+1}} - x_{t_k}), \tag{16}$$

$$\int_t^{t'} a_s(x_s)\cdot\mathrm{d}\overleftarrow{x_s} = \lim_{\max_k |t_{k+1}-t_k|\to 0}\sum_k a_{t_{k+1}}(x_{t_{k+1}})\cdot(x_{t_{k+1}} - x_{t_k}). \tag{17}$$

Note that this is different from the Riemann Integral, where the "direction" of summation does not matter and needs to converge to the same value.

Similarly, for the following forward and backward CTMCs:

$$\partial_t p_t = \mathrm{M}_t^\top p_t, \quad \partial_t p_t = -\mathrm{N}_t^\top p_t \tag{18}$$

we have

$$R_{t,t'}(x_{[t,t']}) = \exp\left(\int_t^{t'}\left[\mathrm{N}_s(x_s, x_s) - \mathrm{M}_s(x_s, x_s)\right]\mathrm{d}s + \sum_{s:X_s^- \neq x_s}\log\frac{\mathrm{N}_s(x_s^-, x_s)}{\mathrm{M}_s(x_s, x_s^-)}\right). \tag{19}$$

## A.3 CHOICE OF $\mathbb{Q}$ PROCESSES

In this section, we exemplify some choices for the proposal process $\overrightarrow{\mathbb{Q}}$ and $\overleftarrow{\mathbb{Q}}'$. Note that none of these choices is unique. In fact, one enjoys large flexibility in choosing these dynamics. Also, for fair comparison, we use the same proposal for SMC with RNE (He et al., 2025c).

**Tempering:** in the Gaussian diffusion (with forward and backward drifts $f_t$ and $g_t$ as defined in Equations (5) and (6)), one may choose the backward drift to be the standard noising kernel $b_t = f_t$, and the forward drift as $a_t = f_t - \beta\sigma_t^2\nabla\log p_t$ following (He et al., 2025c; Skreta et al., 2025).

**Reward-tilting:** we can also choose the backward drift to be the standard noising kernel $b_t = f_t$, and the forward drift with a reward-guidance as $a_t = f_t - \sigma_t^2(\nabla\log p_t + \nabla r_t)$ (Dou & Song, 2024). When the reward is non-differentiable or expensive to compute, we can also set $a_t = f_t - \sigma_t^2\nabla\log p_t$. In this case, we will rely entirely on the SMC/PT correction.

**Composition:** we can choose the backward drift to be the standard noising kernel $b_t = f_t$, and the forward drift using the summation of scores as $a_t = f_t - \sigma_t^2 (\sum_j \nabla \log p_t^j)$ similar to He et al. (2025c); Skreta et al. (2025).

**CFG Debiasing:** CFG debiasing is a simple combination of annealing and composition. However, as it's commonly adopted in diffusion models, we also discuss it as an individual case. For CFG debiasing, we can choose the backward drift to be the standard noising kernel $b_t = f_t$, and the forward drift using the standard CFG dynamics as $a_t = f_t - \sigma_t^2 (w \nabla \log p_t^j(\cdot) + (1 - w) \nabla \log p_t^j(\cdot|c))$. We can also use a different CFG strength $w'$ for the proposal drift $a_t = f_t - \sigma_t^2 (w' \nabla \log p_t^j(\cdot) + (1 - w') \nabla \log p_t^j(\cdot|c))$, as considered by Lee et al. (2025).

**CTMC CFG Debiasing:** for CFG debiasing in CTMC, we can also follow the heuristics proposed by Lee et al. (2025). Where we set $A_t = \Lambda_t$ as the simple masking process; and we set $B_t(x, y) = \Lambda_t(y, x)(\frac{p_t(y|c)}{p_t(x|c)})^w (\frac{p_t(y)}{p_t(x)})^{1-w}$ where the pretrained conditional and unconditional model provide us the conditional and unconditional concrete score $\frac{p_t(y|c)}{p_t(x|c)}$ and $\frac{p_t(y)}{p_t(x)}$. Similar to Gaussian diffusion, we can choose a different strength $w'$ for the proposal process, as adopted by Lee et al. (2025).

## A.4 DISCRETISED FORMULA FOR PT CONTROL

In the main paper, we focus our discussion on the continuous-time formula. In practice, both the simulation of the processes and the calculation of the acceptance rate need to be discretised in time. In this section, we will discuss this discrete form.

### A.4.1 GAUSSIAN DIFFUSION

**Simulation with Euler–Maruyama discretisation**  For the diffusion models, we will apply EM discretisation for the forward and backward dynamics. Note that this is not the only choice. We can also use the DDPM transition kernel (Ho et al., 2020) or the exponential integrator.

$$\text{Discretised Forward proposal:} \quad \overrightarrow{X}_{s+\delta t} = \overrightarrow{X}_s + a_s(\overrightarrow{X}_s)\delta t + \sigma_s \sqrt{\delta t}\epsilon, \quad \epsilon \sim \mathcal{N}(0, \mathbf{I}), \quad (20)$$

$$\text{Discretised Backward proposal:} \quad \overrightarrow{X}'_{s-\delta t} = \overleftarrow{X}'_s - b_t(\overleftarrow{X}'_s)\delta t + \sigma_s \sqrt{\delta t}\epsilon', \quad \epsilon' \sim \mathcal{N}(0, \mathbf{I}), \quad (21)$$

**Acceptance rate with discrete kernel**  With the same discretisation, we can calculate the $R$ with Gaussian densities: concretely, consider the forward and backward processes discretised into $K$ steps: $t = t_0 < t_1 < \cdots < t_K = t'$.

$$\hat{R}_{t,t'}^{\mathbb{Q}}(x_{[t,t']}) = \frac{\prod_{k=1}^K \overleftarrow{B}_{t_{k-1}|t_k}^{\mathbb{Q}}(x_{t_{k-1}}|x_{t_k})}{\prod_{k=1}^K \overrightarrow{F}_{t_k|t_{k-1}}^{\mathbb{Q}}(x_{t_k}|x_{t_{k-1}})}. \quad (22)$$

where

$$\overleftarrow{B}_{t_{k-1}|t_k}^{\mathbb{Q}}(x_{t_{k-1}}|x_{t_k}) = \mathcal{N}(x_{t_{k-1}}; x_{t_k} - b_{t_k}(x_{t_k})|t_k - t_{k-1}|, \sigma_t^2|t_k - t_{k-1}|) \quad (23)$$

$$\overrightarrow{F}_{t_{k+1}|t_k}^{\mathbb{Q}}(x_{t_{k+1}}|x_{t_k}) = \mathcal{N}(x_{t_{k+1}}; x_{t_k} + a_{t_k}(x_{t_k})|t_{k+1} - t_k|, \sigma_t^2|t_{k+1} - t_k|) \quad (24)$$

$\hat{R}_{t,t'}^{\mathbb{P}}$ follows the same calculation. For the diffusion model in Equations (5) and (6), we have

$$\hat{R}_{t,t'}^{\mathbb{P}}(x_{[t,t']}) = \frac{\prod_{k=1}^K \overleftarrow{B}_{t_{k-1}|t_k}^{\mathbb{P}}(x_{t_{k-1}}|x_{t_k})}{\prod_{k=1}^K \overrightarrow{F}_{t_k|t_{k-1}}^{\mathbb{P}}(x_{t_k}|x_{t_{k-1}})}. \quad (25)$$

where

$$\overleftarrow{B}_{t_{k-1}|t_k}^{\mathbb{P}}(x_{t_{k-1}}|x_{t_k}) = \mathcal{N}(x_{t_{k-1}}; x_{t_k} - g_{t_k}(x_{t_k})|t_k - t_{k-1}|, \sigma_t^2|t_k - t_{k-1}|) \quad (26)$$

$$\overrightarrow{F}_{t_{k+1}|t_k}^{\mathbb{P}}(x_{t_{k+1}}|x_{t_k}) = \mathcal{N}(x_{t_{k+1}}; x_{t_k} + f_{t_k}(x_{t_k})|t_{k+1} - t_k|, \sigma_t^2|t_{k+1} - t_k|) \quad (27)$$

**Reference process to stabilise the calculation (He et al., 2025c)**  Using the above discretisation directly can lead to numerical issues. This issue was observed and analysed in He et al. (2025c), arising from the misalignment of variance between forward and backward kernels. To address this,

He et al. (2025c) introduce an analytical reference to convert forward-backward kernel ratios into forward-forward and backward-backward kernel ratios.

Concretely, we introduce a reference diffusion process:

$$\overrightarrow{Y}_0 \sim \gamma_0, \quad \overrightarrow{Y}_s \sim \gamma_s, \quad \mathrm{d}\overrightarrow{Y}_s = f_s(\overrightarrow{Y}_s)\,\mathrm{d}s + \sigma_s\,\mathrm{d}\overrightarrow{W}_s. \tag{28}$$

where $\gamma_0$ is chosen to be a Gaussian. Therefore, all $\gamma_t$ are Gaussian with tractable mean and variance. We can hence write down its time-reversal easily:

$$\overleftarrow{Y}_1 \sim \gamma_1, \quad \overleftarrow{Y}_s \sim \gamma_s, \quad \mathrm{d}\overleftarrow{Y}_s = h_s(\overleftarrow{Y}_s)\,\mathrm{d}s + \sigma_s\,\mathrm{d}\overleftarrow{W}_s, \tag{29}$$

where $h_s = f_s - \sigma_s^2 \nabla \log \gamma_s$. We can also calculate the RNE for this reference process, which we denote by $R_{t,t'}^\Gamma$, and following Equation (4), we know

$$\frac{\gamma_{t'}(x_{t'})}{\gamma_t(x_t)} R_{t,t'}^\Gamma(x_{[t,t']}) = 1. \tag{30}$$

In practice we calculate $R_{t,t'}^\Gamma(x_{[t,t']})$ as:

$$\hat{R}_{t,t'}^\Gamma(x_{[t,t']}) = \frac{\prod_{k=1}^K \overleftarrow{B}_{t_{k-1}|t_k}^\Gamma(x_{t_{k-1}}|x_{t_k})}{\prod_{k=1}^K \overrightarrow{F}_{t_k|t_{k-1}}^\Gamma(x_{t_k}|x_{t_{k-1}})}. \tag{31}$$

$$\overleftarrow{B}_{t_{k-1}|t_k}^\Gamma(x_{t_{k-1}}|x_{t_k}) = \mathcal{N}(x_{t_{k-1}}; x_{t_k} - h_{t_k}(x_{t_k})|t_k - t_{k-1}|, \sigma_t^2|t_k - t_{k-1}|) \tag{32}$$

$$\overrightarrow{F}_{t_{k+1}|t_k}^\Gamma(x_{t_{k+1}}|x_{t_k}) = \mathcal{N}(x_{t_{k+1}}; x_{t_k} + f_{t_k}(x_{t_k})|t_{k+1} - t_k|, \sigma_t^2|t_{k+1} - t_k|) \tag{33}$$

We can see both $\hat{R}_{t,t'}^\Gamma$ and $\hat{R}_{t,t'}^\mathbb{P}$ (or $\hat{R}_{t,t'}^\mathbb{Q}$) have the form of forward-backward kernel ratios; hence, dividing them yields a conversion from forward-backward to forward-forward and backward-backward kernel ratios. More precisely, we calculate $\hat{R}_{t,t'}^\mathbb{P}$ and $\hat{R}_{t,t'}^\mathbb{Q}$ as follows:

$$\hat{R}_{t,t'}^\mathbb{Q}(x_{[t,t']}) = \frac{\prod_{k=1}^K \overleftarrow{B}_{t_{k-1}|t_k}^\mathbb{Q}(x_{t_{k-1}}|x_{t_k})}{\prod_{k=1}^K \overleftarrow{B}_{t_{k-1}|t_k}^\Gamma(x_{t_{k-1}}|x_{t_k})} \frac{\prod_{k=1}^K \overrightarrow{F}_{t_k|t_{k-1}}^\Gamma(x_{t_k}|x_{t_{k-1}})}{\prod_{k=1}^K \overrightarrow{F}_{t_k|t_{k-1}}^\mathbb{Q}(x_{t_k}|x_{t_{k-1}})} \frac{\gamma_t(x_t)}{\gamma_{t'}(x_{t'})}. \tag{34}$$

In fact, we emperically observed that CREPE remains robust than SMC even without the use of a reference. This is because, unlike the SMC weights, the PT acceptance ratio always involves ratios of the $R$'s, which naturally mitigate the variance misalignment. Therefore, we only employ the reference in inference-time tempering experiments, where we found it yields better results.

### A.4.2 CTMC

**Simulation with Euler discretisation**  For CTMCs, we will apply Euler discretisation for the forward and backward dynamics. More precisely, for each token in the sample, we have

$$\text{Discretised Forward proposal:} \quad \overrightarrow{p}(x_{s+\delta t}|x_s) = \delta_{x_{s+\delta t}, x_s} + A_s(x_s, x_{s+\delta t})\delta t + o(\delta t), \tag{35}$$

$$\text{Discretised Backward proposal:} \quad \overleftarrow{p}(x'_{s-\delta t}|x'_s) = \delta_{x'_{s-\delta t}, x'_s} + B_t(x'_s, x'_{s-\delta t})\delta t + o(\delta t) \tag{36}$$

where $\delta_{x_{t+\delta t}, x_t}$ denotes the delta function, which equals 1 when $x_{t+\delta t} = x_t$ and 0 otherwise. This defines a categorical distribution. In practice, we ignore the $o(\delta t)$ term. After evaluating the probabilities for all categories, we clip them to be non-negative and then renormalise.

**Acceptance rate with discrete kernel**  We calculate $R$-s with the same formula as Equations (22) and (25). The only difference is that the transition kernels are defined by Categorical probability instead of Gaussian densities. Assuming the codebook size is $V$, for each token in the mask diffusion defined in Equations (7) and (8), we have

$$\overleftarrow{B}_{t_{k-1}|t_k}^\mathbb{P}(x_{t_{k-1}}|x_{t_k}) = \text{Cat}(x_{t_{k-1}}|[\overleftarrow{p}_1, \overleftarrow{p}_2, ..., \overleftarrow{p}_V]^\mathbb{P}(x_{t_k})) \tag{37}$$

$$\overrightarrow{F}_{t_{k+1}|t_k}^\mathbb{P}(x_{t_{k+1}}|x_{t_k}) = \text{Cat}(x_{t_{k+1}}|[\overrightarrow{p}_1, \overrightarrow{p}_2, ..., \overrightarrow{p}_V]^\mathbb{P}(x_{t_k})) \tag{38}$$

where $[\overrightarrow{p}_1, \overrightarrow{p}_2, ..., \overrightarrow{p}_V]^{\mathbb{P}}(x_{t_k})$ represents the probability vector for $x_{t_{k+1}} = [v_1, \cdots, v_V]$, with each element given by

$$[\overrightarrow{p}_1, \overrightarrow{p}_2, ..., \overrightarrow{p}_V]^{\mathbb{P}}(x_{t_k}) = \begin{bmatrix} \delta_{v_1, x_{t_k}} + \Lambda_{t_k}(x_{t_k}, v_1)\delta t \\ \delta_{v_2, x_{t_k}} + \Lambda_{t_k}(x_{t_k}, v_2)\delta t \\ \cdots \\ \delta_{v_V, x_{t_k}} + \Lambda_{t_k}(x_{t_k}, v_V)\delta t \end{bmatrix} \tag{39}$$

and:

$$[\overleftarrow{p}_1, \overleftarrow{p}_2, ..., \overleftarrow{p}_V]^{\mathbb{P}}(x_{t_k}) = \begin{bmatrix} \delta_{v_1, x_{t_k}} + \Lambda'_{t_k}(x_{t_k}, v_1)\delta t \\ \delta_{v_2, x_{t_k}} + \Lambda'_{t_k}(x_{t_k}, v_2)\delta t \\ \cdots \\ \delta_{v_V, x_{t_k}} + \Lambda'_{t_k}(x_{t_k}, v_V)\delta t \end{bmatrix} \tag{40}$$

We also remove `nan` and `inf` values, and clip all values to be larger than `1e-8`. This makes the sum of all elements deviate from 1, but we found it to work well in practice, as when $\delta t$ is small, the deviation is negligible. This setup was also adopted in previous SMC works (Lee et al., 2025).

Similarly,

$$\overleftarrow{B}^{\mathbb{Q}}_{t_{k-1}|t_k}(x_{t_{k-1}}|x_{t_k}) = \text{Cat}(x_{t_{k-1}}|[\overleftarrow{p}_1, \overleftarrow{p}_2, ..., \overleftarrow{p}_V]^{\mathbb{Q}}(x_{t_k})) \tag{41}$$

$$\overrightarrow{F}^{\mathbb{Q}}_{t_{k+1}|t_k}(x_{t_{k+1}}|x_{t_k}) = \text{Cat}(x_{t_{k+1}}|[\overrightarrow{p}_1, \overrightarrow{p}_2, ..., \overrightarrow{p}_V]^{\mathbb{Q}}(x_{t_k})) \tag{42}$$

and

$$[\overrightarrow{p}_1, \overrightarrow{p}_2, ..., \overrightarrow{p}_V]^{\mathbb{Q}}(x_{t_k}) = \begin{bmatrix} \delta_{v_1, x_{t_k}} + A_{t_k}(x_{t_k}, v_1)\delta t \\ \delta_{v_2, x_{t_k}} + A_{t_k}(x_{t_k}, v_2)\delta t \\ \cdots \\ \delta_{v_V, x_{t_k}} + A_{t_k}(x_{t_k}, v_V)\delta t \end{bmatrix} \tag{43}$$

and:

$$[\overleftarrow{p}_1, \overleftarrow{p}_2, ..., \overleftarrow{p}_V]^{\mathbb{Q}}(x_{t_k}) = \begin{bmatrix} \delta_{v_1, x_{t_k}} + B_{t_k}(x_{t_k}, v_1)\delta t \\ \delta_{v_2, x_{t_k}} + B_{t_k}(x_{t_k}, v_2)\delta t \\ \cdots \\ \delta_{v_V, x_{t_k}} + B_{t_k}(x_{t_k}, v_V)\delta t \end{bmatrix} \tag{44}$$

Also, we do not use the reference process as we do not observe an instability issue.

## A.5 ACCEPTANCE RATE FOR REWARD, CFG AND COMPOSITION

### A.5.1 TEMPERING

Suppose $\pi_0(x) \propto p_0^j(x)^\beta$ for some $\beta > 0$, with annealing path $\pi_t(x) \propto p_t^j(x)^\beta$ the maringal density ratio satisfies,

$$\frac{\pi_{t'}(x')}{\pi_t(x)} \propto \left(\frac{p_{t'}^j(x')}{p_t^j(x)}\right)^\beta = R_{t,t}^{\mathbb{P}^j}(x_{[t,t']})^{-\beta}, \tag{45}$$

and acceptance function equals,

$$\alpha_{t,t'}(x_{[t,t']}, x'_{[t,t']}) = \min\left\{1, \frac{R_{t,t'}^{\mathbb{P}^j}(x'_{[t,t']})^\beta}{R_{t,t'}^{\mathbb{P}^j}(x_{[t,t']})^\beta} \cdot \frac{R_{t,t'}^{\mathbb{Q}}(x_{[t,t']})}{R_{t,t'}^{\mathbb{Q}}(x'_{[t,t']})}\right\}. \tag{46}$$

### A.5.2 REWARD-TILTING/POSTERIOR SAMPLING

Suppose $\pi_0(x) = p_0^j(x)\exp(r_0(x))$ given a reward/likihood function $r_0^j(x)$. We can construct an annealing path $\pi_t(x) = p_t^j(x)\exp(r_t(x))$, where $r_t$ is a user specified reward function such that coincides with $r_0$ at $t = 0$. A heuristic choice (Wu et al., 2023) is

$$r_t(x) = \gamma_t r_0\left(\mathbb{E}_{X_t^j \sim p_t^j}[X_0^j|X_t^j = x]\right), \tag{47}$$

with boundary conditions $\gamma_1 = 0$ and $\gamma_0 = 1$, where the expectation is calculated with Tweedie's formula (Efron, 2011) with the pretrained diffusion. The marginal density ratio satisfies,

$$\frac{\pi_{t'}(x_{t'})}{\pi_t(x_t)} \propto \frac{p_{t'}^j(x_{t'})}{p_t^j(x_t)} \cdot \frac{\exp(r_{t'}(x_{t'}))}{\exp(r_t(x_t))} = \frac{\exp(r_{t'}(x_{t'}) - r_t(x_t))}{R_{t,t'}^{\mathbb{P}^j}(x_{[t,t']})}, \tag{48}$$

and hence the acceptance probability equals,

$$\alpha_{t,t'}(x_{[t,t']}, x'_{[t,t']}) = \min\left\{1, \frac{\exp(r_{t'}(x_{t'}) - r_t(x_t))}{\exp(r_{t'}(x'_{t'}) - r_t(x'_t))} \frac{R_{t,t'}^{\mathbb{P}^j}(x'_{[t,t']})}{R_{t,t'}^{\mathbb{P}^j}(x_{[t,t']})} \cdot \frac{R_{t,t'}^{\mathbb{Q}}(x_{[t,t']})}{R_{t,t'}^{\mathbb{Q}}(x'_{[t,t']})}\right\}. \tag{49}$$

### A.5.3 MODEL COMPOSITION

When $\pi_0(x) = \prod_{j=1}^J p_0^j(x)$ with annealing path $\pi_0(x) = \prod_{j=1}^J p_t^j(x)$, the marginal density ratio satisfies,

$$\frac{\pi_{t'}(x_{t'})}{\pi_t(x_t)} \propto \prod_{j=1}^J \frac{p_{t'}^j(x_{t'})}{p_t^j(x_t)} = \prod_{j=1}^J R_{t,t'}^{\mathbb{P}^j}(x_{[t,t']})^{-1}, \tag{50}$$

and the acceptance probability equals,

$$\alpha_{t,t'}(x_{[t,t']}, x'_{[t,t']}) = \min\left\{1, \prod_{j=1}^J \frac{R_{t,t'}^{\mathbb{P}^j}(x'_{[t,t']})}{R_{t,t'}^{\mathbb{P}^j}(x_{[t,t']})} \cdot \frac{R_{t,t'}^{\mathbb{Q}}(x_{[t,t']})}{R_{t,t'}^{\mathbb{Q}}(x'_{[t,t']})}\right\}. \tag{51}$$

### A.6 DETAILS ON LOCAL EXPLORATION

In this section, we discuss our choice for local exploration.

### A.6.1 GAUSSIAN DIFFUSION

For Gaussian diffusion, we modify the corrector step in the predictor-corrector algorithm by Song et al. (2021b). More concretely, we apply the Unadjusted Langevin Algorithm (ULA) using the score of $\pi_t$. This is, in most cases, available and is simply the combination of the pretrained diffusion's score $p_t$. One exception is the reward-tilting case. In this case, if the reward is cheap and differentiable, we can simply take the gradient of it to calculate the score of $\pi_t$. In our experiments on trajectory stitching, we apply this local move. On the other hand, when the reward is non-differentiable or expensive to take a gradient, we can omit the local move step, as discussed in the footnote in Section 3.1. For our experiment on prompted reward-tilting, we omit this local move.

We also find that using the step size proposed by Song et al. (2021b) leads to instability in our case. Therefore, we choose to use the step size aligned with the "step size" of the denoising process. Precisely, the standard deviation of the Gaussian noise in the EM step (with discretisation size $\delta t$) for Equation (6) at step $t$ is $\sigma_t \sqrt{\delta t}$. In ULA at $t$, we hence set the step size to $\frac{1}{2}\sigma_t^2 \delta t$ so that the standard deviation of the Gaussian noise added in the ULA aligns with that in the denoising kernel.

### A.6.2 CTMC

For CTMC, the concrete score approximates the density ratio (Lou et al., 2023)

$$s_t^\theta(x)_y \approx \frac{p_t(x)}{p_t(y)}. \tag{52}$$

Using this relation, we can define a rate matrix based on the concrete score that satisfies detailed balance (DB). One choice is a Metropolis-Hastings-style rate matrix:

$$Q(x,y) = r(x,y)\min(1, s_t^\theta(x)_y)\mathbb{1}_{x \neq y} - \mathbb{1}_{x=y}\sum_{y \neq x} Q(x,y), \tag{53}$$

where $r(x, y) \geq 0$ is the proposal kernel. For example, we could use a uniform proposal:

$$r(x, y) = \frac{1}{|\mathbb{X}| - 1}, \tag{54}$$

Another option is to set the proposal kernel to the noising/masking process $r(x, y) = \Lambda_t(x, y)$ directly. However, for tasks such as CFG, we cannot use the predictor-corrector algorithm proposed in Campbell et al. (2022), as we do not have access to the concrete score for the marginal of the CFG dynamics. Therefore, we need to resort to these MH-style correctors. In our experiments, we also found that CTMC also performs well without local moves.

## A.7 COMPUTATIONAL COMPARISON BETWEEN SMC AND CREPE

Here, we include a discussion on the computation cost comparison between SMC and CREPE. To keep the discussion simple, we will use the Gaussian diffusion model as an example.

Assume in SMC, we resample every $K$ steps, and in total we resample $M$ times with a batch size of $N$. In CREPE, we run PT at $M$ diffusion times, and discrete $K$ steps for communication, and in total collect samples with $N$ iterations. In both cases, the number of network evaluations will be aligned.

For SMC, it's easy to see we need $M \cdot K \cdot N$ NFEs in total.

For the CREPE communication step, at each iteration, we only swap half of the levels, but we need to propose and calculate the RND for both directions, contributing to $M \cdot K/2 \times 2$ NFEs. The local move will not need new NFEs, as we already evaluate the score for the newly accepted sample when we calculate the RND. Therefore, in total, CREPE also need $M \cdot K \cdot N$ NFEs.

# B EXPERIMENTAL DETAILS

## B.1 TEMPERING

**Model and Training Details.** For all three molecules, we use EGNN network following (Hoogeboom et al., 2022). We use the VE (EDM) schedule following Karras et al. (2022), and adopt their preconditioning $(c_{in}, c_{out}, c_{skip})$ as well. For Dipeptide and Tetrapeptide, the network has 5 layers, each with 256 hidden units. For Hexapeptide, we increase the network size to 5 layers and 512 hidden units. Networks are trained with Adam with a learning rate `1e-4` until convergence. We also apply an EMA with a rate of 0.999.

**Data.** The samples were gathered following He et al. (2025b) from a 5-microsecond simulation with Generalised Born implicit solvent implemented in `openmmtools` (Chodera et al., 2025) with AMBER ff96 classical force field. The Langevin middle integrator is implemented in Eastman et al. (2023) with a friction of 1/picosecond and a step size of 2 femtoseconds.

**Metrics.** We evaluate energy and interatomic distance TVD following Akhound-Sadegh et al. (2024). We use the implementation of W2 distance by Akhound-Sadegh et al. (2024) as well to evaluate the sample W2. However, our system is invariant to rotation and translation. Therefore, we align all samples to a reference sample by the Kabsch algorithm (Kabsch, 1976) before evaluating W2 distance. We use PyEMMA (Scherer et al., 2015) to project samples into 2D using TICA with `lag=8`. The TICA is fitted on the ground truth trajectory. We first transfer the sample from Cartesian coordinates to backbone torsion angles before applying TICA. We then employ the MMD to the 2D TICA plot, using the implementation by Chen et al. (2024), with a fixed bandwidth chosen to be the mid-distance between ground truth samples data and maintained throughout the evaluation of different methods.

Since some of the metrics are expensive to evaluate on large datasets, we randomly select 5,000 samples from both the ground truth and our generated samples by SMC or CREPE when evaluating all these metrics. We repeat this procedure three times and report the mean along with error bars.

**CREPE Hyperparameters** Our SDE follows EDM (Karras et al., 2022) with forward SDE defined as $\mathrm{d}X_t = \sqrt{2t}\mathrm{d}W_t$. We choose $t \in [t_{\min} = 0.001, t_{\max} = 10]$, and discretised into 201 steps following Karras et al. (2022) by $[t_{\max}^{1/\rho} + \frac{\text{step\_idx}}{200}(t_{\min}^{1/\rho} - t_{\max}^{1/\rho})]^\rho$, with $\rho = 7$. We then select one PT level every 4 diffusion steps, resulting in $M = 51$ levels, with each level containing $K = 4$ steps. We run CREPE for 50,000 iterations, collecting all samples after iteration 1000.

We run CREPE with both communication and local move. We also use the reference process when calculating $R$ as described in Appendix A.4.1.

**SMC (FKC and RNE) Hyperparameters** We adopt the same schedule and discretisation as CREPE. We resample every 4 diffusion denoising steps to align with the setup of CREPE. Both FKC and RNE are run in batches, a common strategy to mitigate the diversity loss in SMC (He et al., 2025c; Lee et al., 2025). Specifically, for each target, we run 50 batches of size 1,000, so that the overall computational budget for SMC and CREPE is aligned.

## B.2 IMAGE CFG DEBIASING

**Model Details** We use the EDM2-S model on ImageNet-64 and the EDM2-XS model on ImageNet-512 (Karras et al., 2024). The ImageNet-64 model is in pixel-space, while the ImageNet-512 model is a latent diffusion model.

**CREPE Hyperparameters** We aim to debias CFG with $w = 1.7$. Our SDE follows EDM (Karras et al., 2022) with forward SDE defined as $dX_t = \sqrt{2t}dW_t$. We choose $t \in [t_{\min} = 0.002, t_{\max} = 80]$, and discretised into 128 steps by $[t_{\max}^{1/\rho} + \frac{\text{step\_idx}}{127}(t_{\min}^{1/\rho} - t_{\max}^{1/\rho})]^\rho$, with $\rho = 7$. However, we only apply PT within the first 100 steps, after which we proceed using standard Euler–Maruyama updates with CFG dynamics. We select one PT level every diffusion step, resulting in $M = 100$ levels, with each level containing $K = 1$ step. We omit local exploration steps.

**SMC (FKC) Hyperparameters** We use the same SDE, together with systematic resampling, adopting the same 128-step EDM schedule with FKC applied until timestep 100. After this, we proceed using standard Euler–Maruyama updates, same as CREPE.

**Evaluation details for Figure 5** In Figure 5, we report IR, CLIP and FID for different numbers of particles $N$ used in CREPE and FKC. For each setup, we run $B$ independent runs with randomly selected $B$ classes. The selection is fixed between CREPE and FKC. We then gather $NB$ samples to evaluate the metrics. To ensure the number of samples is roughly the same, for each different choice of $N$, we set $B = \text{ceil}(5000/N)$.

## B.3 IMAGE REWARD-TILTING

We first debias CFG with $w = 1.3$, together with reward-tilting. The reward is defined with ImageReward with the prompt we provide.

**Model Details** We use the same model as CFG debiasing: the EDM2-S model on ImageNet-64 and the EDM2-XS model on ImageNet-512 (Karras et al., 2024).

**Reward Details** The final reward $r$ is given by ImageReward (Xu et al., 2023) with the prompt we provide. We also multiply the reward value by 100 as the original magnitude is small. The intermediate reward is defined by $r_t(x_t) = \beta_t \mathbb{E}[x_0|x_t]$ where the expectation is calculated by Tweedie's formula using the pretrained score network. The $\beta_t$ is selected to be a smooth interpolant between $\beta_1 = 0$ and $\beta_0 = 1$. We use $\beta_{t_m} = [\beta_1^{1/\rho} + \frac{m}{M}(\beta_0^{1/\rho} - \beta_1^{1/\rho})]^\rho$, with $\rho = 5$ for the PT level corresponding to $t_m$. The correspondence between $m$ and $t_m$ is described in the following paragraph.

**CREPE Hyperparameters** Our SDE follows EDM (Karras et al., 2022) with forward SDE defined as $dX_t = \sqrt{2t}dW_t$. We choose $t \in [t_{\min} = 0.002, t_{\max} = 80]$, and discretised into 64 steps by $[t_{\max}^{1/\rho} + \frac{\text{step\_idx}}{63}(t_{\min}^{1/\rho} - t_{\max}^{1/\rho})]^\rho$, with $\rho = 7$. However, we only apply PT within the first 32 steps, after which we proceed using standard Euler–Maruyama updates with CFG dynamics. We select one PT level every diffusion step, resulting in $M = 32$ levels, with each level containing $K = 1$ step. We omit local exploration steps as the ImageReward is expensive and not implemented in a differentiable way.

**Prompt Details** The prompts and corresponding class indices are as follows:

"a blue balloon", idx 417

"a colorful pinwheel", idx 723

"a green Christmas stocking", idx 496

"a yellow cab with dark background", idx 468

"an empty shopping cart", idx 791

### B.4 MAZE

**Model and Training Details** We use an MLP with 5 layers and 512 hidden units to parameterise the diffusion model. We use the VE (EDM) schedule following Karras et al. (2022), and adopt their preconditioning $(c_{in}, c_{out}, c_{skip})$ as well. Networks are trained with Adam with a learning rate `1e-4` until convergence. We also apply an EMA with a rate of 0.999.

**Data** We use the `pointmaze-giant-stitch-v0` dataset from Park et al. (2024), which consists of short trajectories of length 64 in a large 2D maze. We visualise some training trajectories in Figure 6 (Left). We follow Luo et al. (2025) to first normalise the data to [-1, 1] for training. When evaluating and visualising, we unnormalise the trajectory back to its original scale.

**Metrics** We evaluate the success rate from the initial point to the target point. (Luo et al., 2025) considers a trajectory successfully navigating through the maze when the distance between the first point in the trajectory and the initial point is smaller than 0.45 (unnormalised scale). However, in our case, we impose a harsher criterion: we additionally require the distance between the tail of the last short trajectory and the head of the next short trajectory to be smaller than 0.45. We impose this criterion as our stitching is performed via a reward function instead of directly training the diffusion model conditional on both ends. Additionally, we also require that any points along the entire stitched trajectory have no overlap with the walls.

**Choice of Reward Function** We first define the reward $r$ for the data in the clean space ($t = 0$), and then provide the formula for the reward $r_t$ when $t > 0$. Recall we want to define a reward function so that: (1) the starting point of the first trajectory is sufficiently close to the initial state, (2) the endpoint of the last trajectory is sufficiently close to the target state, and (3) consecutive trajectories are connected in a tail-to-head manner. We can impose the $L^2$ distance for each of the constraints. But it is known that $L^2$ distance is making an implicit Gaussian noise assumption, which is typically not "sharp" enough. Instead, we can also impose the $L^1$ distance. However, the $L^1$ distance can be too weak when two points are far apart, as it implicitly assumes a Laplacian noise, which may lead to heavy-tailed behaviour. Therefore, to make use of both advantages of $L^1$ and $L^2$, we take the summation of both. We use $X^{j,i}$ to represent the $i$-th point in the $j$-th short trajectory. Also, we use $-1$ to represent the last element, following the index convention of Python. We use $O$ and $P$ to represent the initial and target points.

$$\text{Reward for initial point: } r^O = -\lambda_O(\lambda_{L^2}||X^{0,0} - O||_2^2 + \lambda_{L^1}||X^{0,0} - O||_1^1) \tag{55}$$

$$\text{Reward for target point: } r^P = -\lambda_P(\lambda_{L^2}||X^{-1,-1} - P||_2^2 + \lambda_{L^1}||X^{-1,-1} - P||_1^1) \tag{56}$$

$$\text{Reward for neighboring trajectories:} \tag{57}$$

$$r^N = -\sum_j \lambda_N(\lambda_{L^2}||X^{j,-1} - X^{j+1,0}||_2^2 + \lambda_{L^1}||X^{j,-1} - X^{j+1,0}||_1^1) \tag{58}$$

where we set $\lambda_O = \lambda_P = 100 \times J$, $\lambda_N = 100$, $\lambda_{L^2} = 1$ and $\lambda_{L^2} = 10$. and the final reward is:

$$r = r^O + r^P + r^N \tag{59}$$

For the case where we introduce an intermediate point $I$, we additionally impose

$$r^I = -\sum_i \sum_j \lambda_I \alpha_{ij}(\lambda_{L^2}||X^{j,i} - I||_2^2 + \lambda_{L^1}||X^{j,i} - I||_1^1) \tag{60}$$

where we set $\lambda_I = 100 \times J$ and $\alpha_{ij}$ is an "attention" defined by

$$\alpha_{ij} = \frac{\exp(-\tau\lambda_{L^2}||X^{j,i} - I||_2^2 - \tau\lambda_{L^1}||X^{j,i} - I||_1^1)}{\sum_i \sum_j \exp(-\tau\lambda_{L^2}||X^{j,i} - I||_2^2 - \tau\lambda_{L^1}||X^{j,i} - I||_1^1)} \tag{61}$$

and the temperature $\tau = 10$. The final results of CREPE will not be strongly influenced by the hyperparameter choices of the reward. However, there are two main principles to tune these values: (1) the reward strength should not be too small, as the trajectory will not be well-connected; (2) the reward strength should not be too large, otherwise the PT swap rate will be 0 at some PT levels.

Therefore, when tuning these hyperparameters, one may not need to run the algorithm for a long time. If the short trajectories do not form a tail-to-head manner quickly, then the strength needs to be increased. On the other hand, if the communication rate is always 0 at some PT levels, the strength needs to be reduced.

We then consider how to choose the intermediate reward $r_t$ at diffusion time step $t$. We define it following the same principle as Chung et al. (2023):

$$r_t(X_t^0, X_t^1, \cdots, X_t^J) = \beta_t \cdot r_t(\mathbb{E}[X_0|X_t^0], \mathbb{E}[X_0|X_t^1], \cdots, \mathbb{E}[X_0|X_t^J]) \tag{62}$$

where we use $X_t^j$ to represent the $j$-th short trajectory and $\mathbb{E}[X_0|X_t^j]$ are calculated by Tweedie's formula using the learned score network. The $\beta_t$ is selected to be a smooth interpolant between $\beta_1 = 0$ and $\beta_0 = 1$. We use $\beta_{t_m} = [\beta_1^{1/\rho} + \frac{m}{M}(\beta_0^{1/\rho} - \beta_1^{1/\rho})]^\rho$, with $\rho = 10$ for the PT level corresponding to $t_m$. In the next paragraph, we describe the correspondence between $m$ and $t_m$.

**CREPE Hyperparameters** Our SDE follows EDM (Karras et al., 2022) with forward SDE defined as $dX_t = \sqrt{2t}dW_t$. We choose $t \in [t_{\min} = 0.001, t_{\max} = 20]$, and discretised into 601 steps following Karras et al. (2022) by $[t_{\max}^{1/\rho} + \frac{\text{step\_idx}}{600}(t_{\min}^{1/\rho} - t_{\max}^{1/\rho})]^\rho$, with $\rho = 7$. We then select one PT level every diffusion step, resulting in $M = 601$ levels, with each level containing $K = 1$ steps. We run CREPE with both communication and local move.

## B.5 CTMC

**Model Details** We follow the experimental setup of Lee et al. (2025) for CTMC experiments on MNIST. We use Campbell et al. (2022)'s UNet with `num_scales=3`, `num_res_blocks=3`, `ch_mult=[1, 2, 4]`, `class_embed_dim=32`, `ch=64`.

**SMC Hyperparameters** We aim to sample from the target with CFG strength $w = 1.2$. We follow (Lee et al., 2025) to perform partial resampling, using a resampling fraction of 80% and an effective size threshold of 0.2 (i.e., trigger resampling when ESS $\leq 0.2$). We discretise the diffusion time horizon with 200 steps, and perform SMC with a batch size of 128. When collecting data for evaluating FID, we repeat 4 batches for each class.

**CREPE Hyperparameters** We use the same setup for $w$ and discretisation steps as SMC. We treat each diffusion step as one PT level, resulting in $M = 200$ levels, with each level containing $K = 1$ steps. We run 512 steps for each class to align the budget with SMC. We did not perform the local move and found it works well.

## B.6 TEXT CTMC

**Task** We consider sentiment-controlled text generation using discrete diffusion. The goal is to generate text with a specified sentiment (positive or negative) by conditioning on the prompt "The sentiment of the text is {negative|positive}".

**Model Details** We finetune the SEDD-Medium model (Lou et al., 2023) for 5 epochs on the SST-2 sentiment control dataset from Amini et al. (2024). The finetuned model serves as the conditional diffusion model for both SMC and CREPE.

**Sampling Setup** We discretise the CTMC process into 100 steps and generate 1000 samples of length 1024. For SMC, we use a batch size of 50 particles. For CREPE, we run 150 burn-in steps followed by 1,000 sampling steps, yielding 1000 samples after burning in. We sweep over target strength $\in \{1.0, 1.2, 1.4, 1.8\}$ and proposal strength $\in \{1.0, 1.2, 1.4, 1.8\}$.

**Evaluation** We evaluate sample quality along two axes:

- **Perplexity**: We finetune a GPT2-XL model with LoRA on the SST-2 dataset and use it to compute perplexity of the generated samples.

- **Sentiment accuracy**: We classify generated samples using a pretrained DistilBERT classifier finetuned for sentiment analysis.[2]

---

[2] https://huggingface.co/Kai1014/distilbert-finetuned

# C ADDITIONAL RESULTS

## C.1 SAMPLES OF DEBIASING CFG ON IMAGENET

Here we provide more examples for CFG Debiasing with FKC and CREPE in Figures 11 to 14. The x-axis is the number of particles: in SMC (e.g., FKC), they are running in parallel, while in CREPE, they are running sequentially. For a clear visualisation, we thin along this axis by a factor of 8. The y-axis corresponds to the diffusion steps: in SMC, they are running sequentially along one direction; while in CREPE, they are running in parallel and undergo mutual communication steps. For a clear visualisation, we thin along this axis by a factor of 2. We can see that SMC tend to have low sample diversity due to repeatedly resampling, while CREPE maintains higher sample diversity after burn-in.

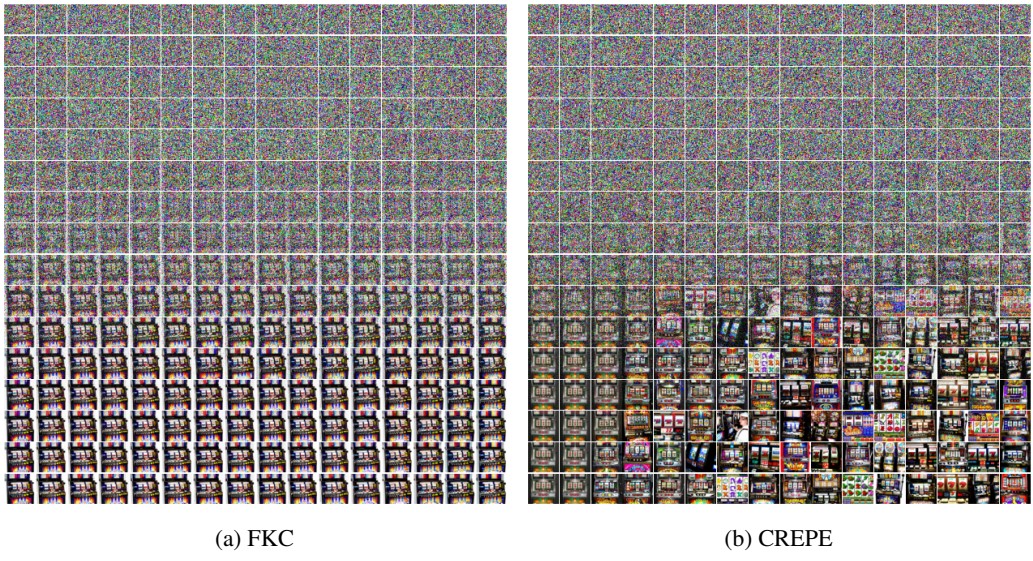

(a) FKC                             (b) CREPE

Figure 11: CFG Debiasing with FKC and CREPE for class "slot" (idx 800).

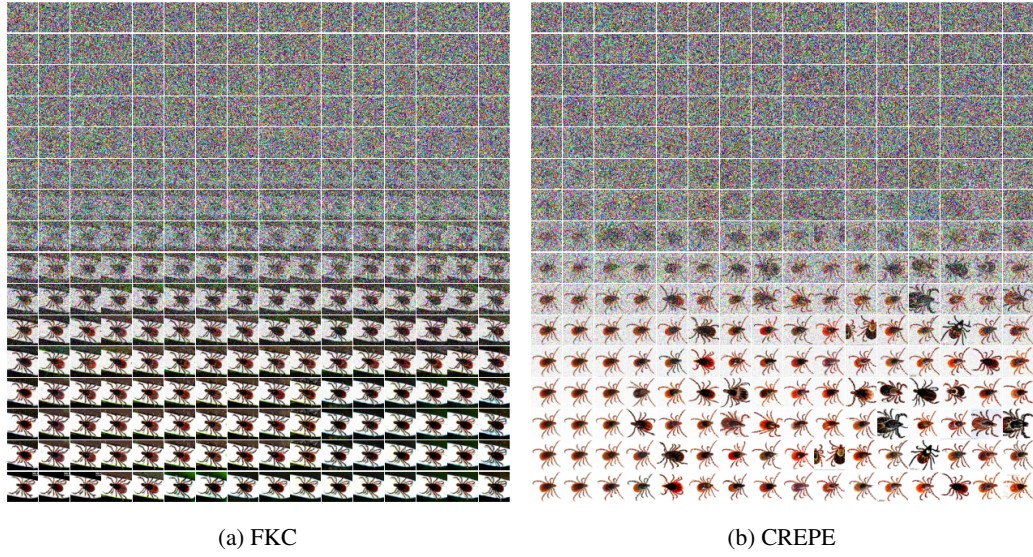

(a) FKC                             (b) CREPE

Figure 12: CFG Debiasing with FKC and CREPE for class "tick" (idx 78).

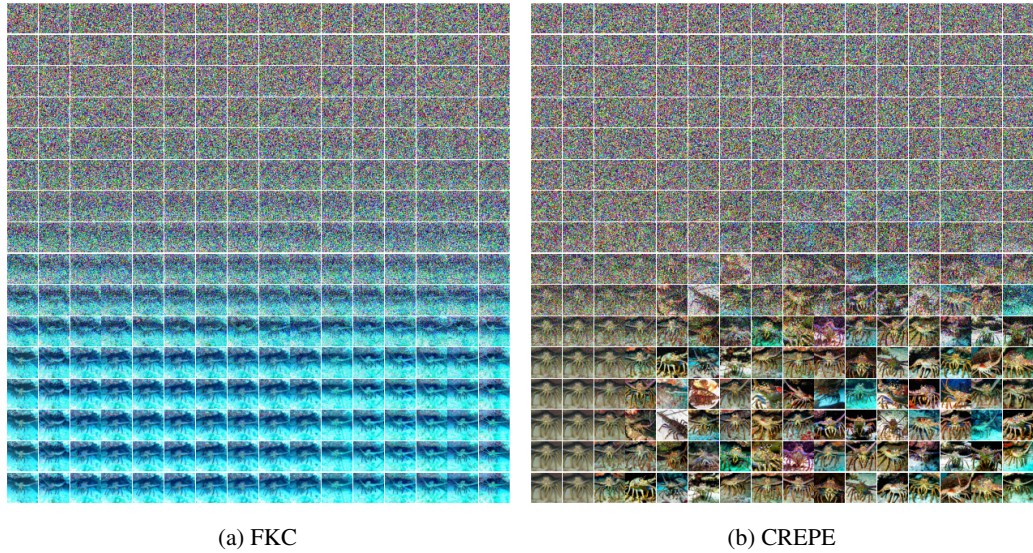

(a) FKC            (b) CREPE

Figure 13: CFG Debiasing with FKC and CREPE for class "spiny lobster" (idx 123).

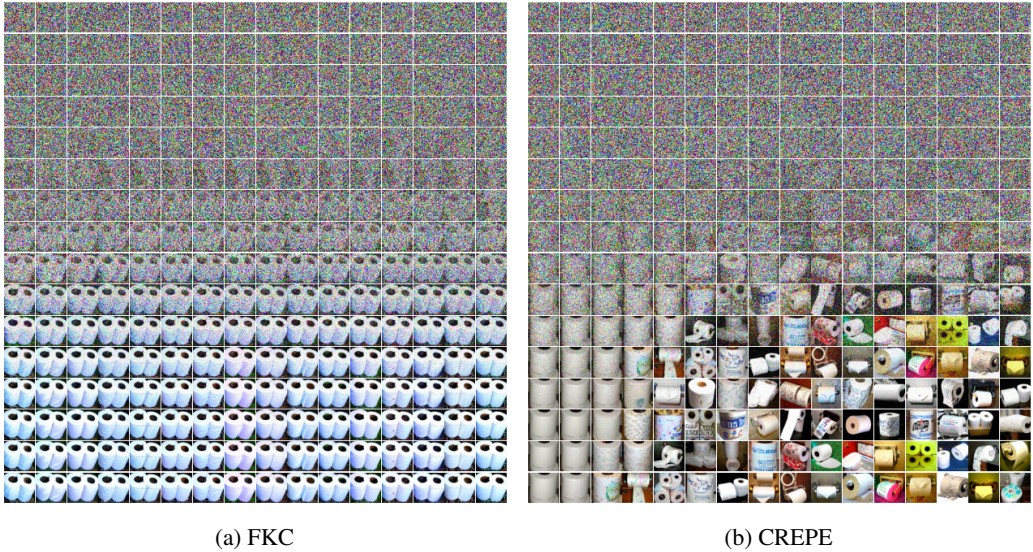

(a) FKC            (b) CREPE

Figure 14: CFG Debiasing with FKC and CREPE for class "toilet tissue" (idx 999).

## C.2 SAMPLES OF REWARD-TILTING ON IMAGENET

In Figure 1, we visualise chains of thinned samples for different prompts. Here, we present the results for the entire PT iteration. As each task has 200 images of $512 \times 512$, we downsize them here. We observe that CREPE produces diverse samples across iterations, aligning with the prompt after the initial burn-in period. We also notice that neighbouring iterations often yield similar images, which arises from the use of non-reversible PT: the last PT level is updated only in alternating (odd or even) iterations.

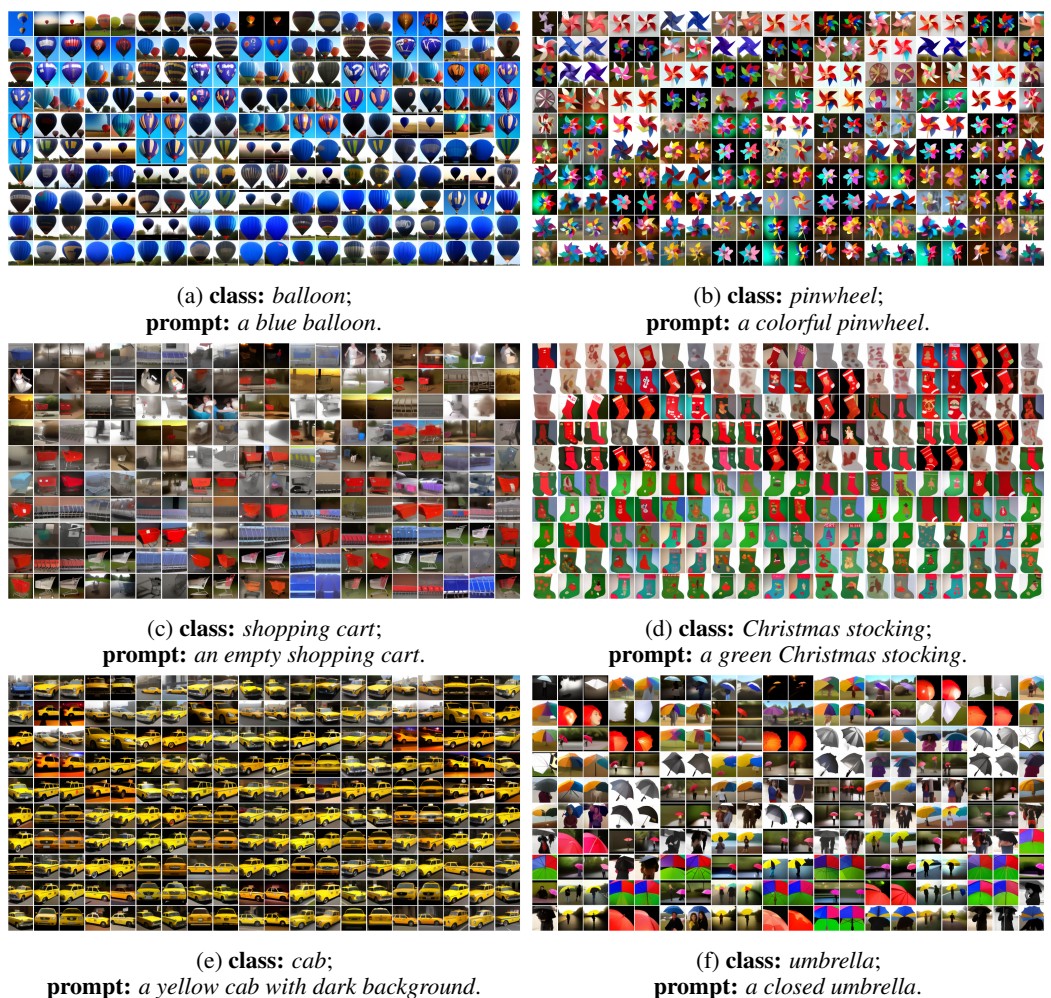

(a) **class:** *balloon*;
**prompt:** *a blue balloon.*

(b) **class:** *pinwheel*;
**prompt:** *a colorful pinwheel.*

(c) **class:** *shopping cart*;
**prompt:** *an empty shopping cart.*

(d) **class:** *Christmas stocking*;
**prompt:** *a green Christmas stocking.*

(e) **class:** *cab*;
**prompt:** *a yellow cab with dark background.*

(f) **class:** *umbrella*;
**prompt:** *a closed umbrella.*

Figure 15: Prompted reward-tilting on ImageNet-512. From left to right and top to bottom, each image corresponds to one CREPE iteration. We visualise the entire image trajectory in the first 200 PT iterations. The last example 15f is a failure mode. Please refer to Appendix C.3 for a discussion on the failure mode.

## C.3 FAILURE MODE OF REWARD-TILTING ON IMAGENET

While we demonstrated that prompted reward-tilting can be used to control image content in finer detail, it does not always succeed. One failure case arises when the class contains no samples that satisfy the prompt. For example, in Figure 15f, the algorithm fails to generate a closed umbrella.

## C.4 SUCCESS RATE OF TRAJECTORY STITCHING

Table 3: Success rates of CREPE across 5 tasks. The CompDiffuser results are taken from Luo et al. (2025), while the CREPE results are averaged over 250 samples within each iteration range.

| Method | | Success rate (%) |
|---|---|---|
| CompDiffuser (Luo et al., 2025) | | 68 |
| CREPE | iteration 0–50k | 8.5 |
| | iteration 50–100k | 59.7 |
| | iteration 100–150k | **84.6** |

## C.5 MORE RESULTS ON TRAJECTORY STITCHING WITH ONLINE REFINEMENT

In Figure 6, we select representative PT iterations to visualise the trajectory samples. To further quantify the performance of online refinement, we evaluate the success rate of reaching the final target and the pass rate through the intermediate point, shown in Figure 16. At iteration 100k, we introduce an additional reward corresponding to the intermediate point. As can be seen, the pass rate through the intermediate point quickly increases after the new reward is added, while the success rate slightly drops but remains high. This demonstrates that CREPE is capable of flexibly incorporating new constraints during sampling and adapting the trajectories online without retraining.

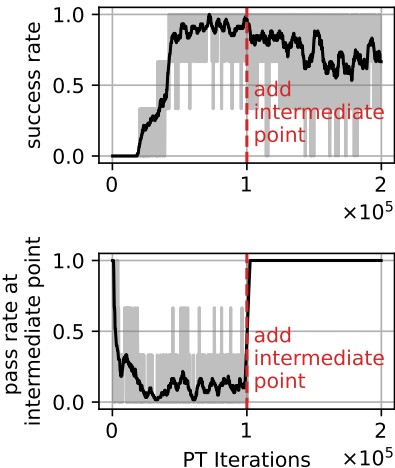

Figure 16: Success rate and pass rate through intermediate point in the trajectory stitching task with online refinement.

# D DEBIASING CFG FOR CTMC ON TEXT

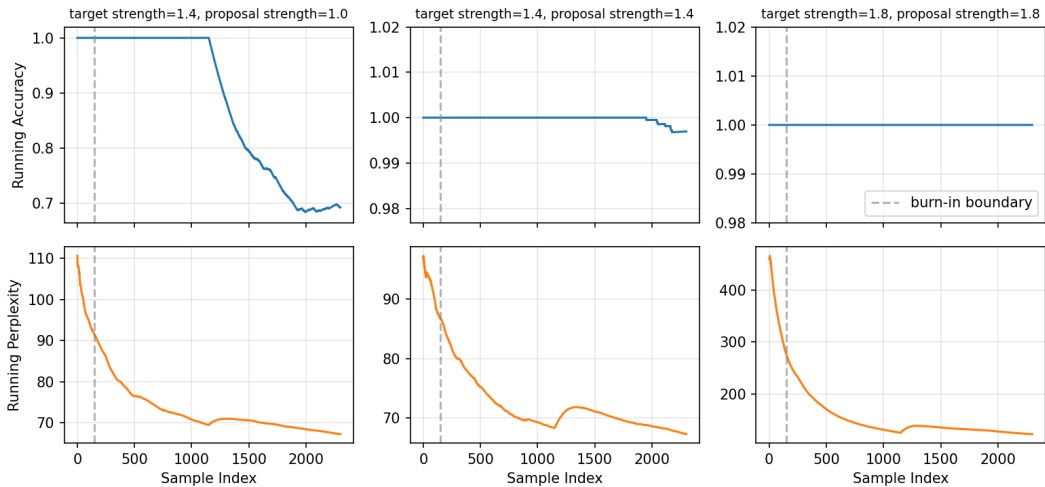

Figure 17: Running average perplexity and sentiment accuracy as a function of CREPE sampling step for three temperature combinations of proposal and target CFG strength.

Figure 17 shows the running average of perplexity and accuracy as CREPE generates samples, for three temperature configurations: (target strength = 1.4, proposal strength = 1.0), (target strength = 1.4, proposal strength = 1.4), and (target strength = 1.8, proposal strength =

1.8). Perplexity decreases as more samples are generated, and accuracy is maintained close to 1.0, indicating effective sentiment control. This demonstrates that CREPE successfully debiases the bias introduced by CFG, leading to a better text sample with lower perplexity. The exception is the $(1.4, 1.0)$ configuration, where accuracy drops around step 1200. A possible explanation is that a bad sample was proposed and accepted at this iteration, revealing a potential weakness of CREPE that requires future exploration.

## D.1 DEBIASING CFG FOR CTMC ON MNIST

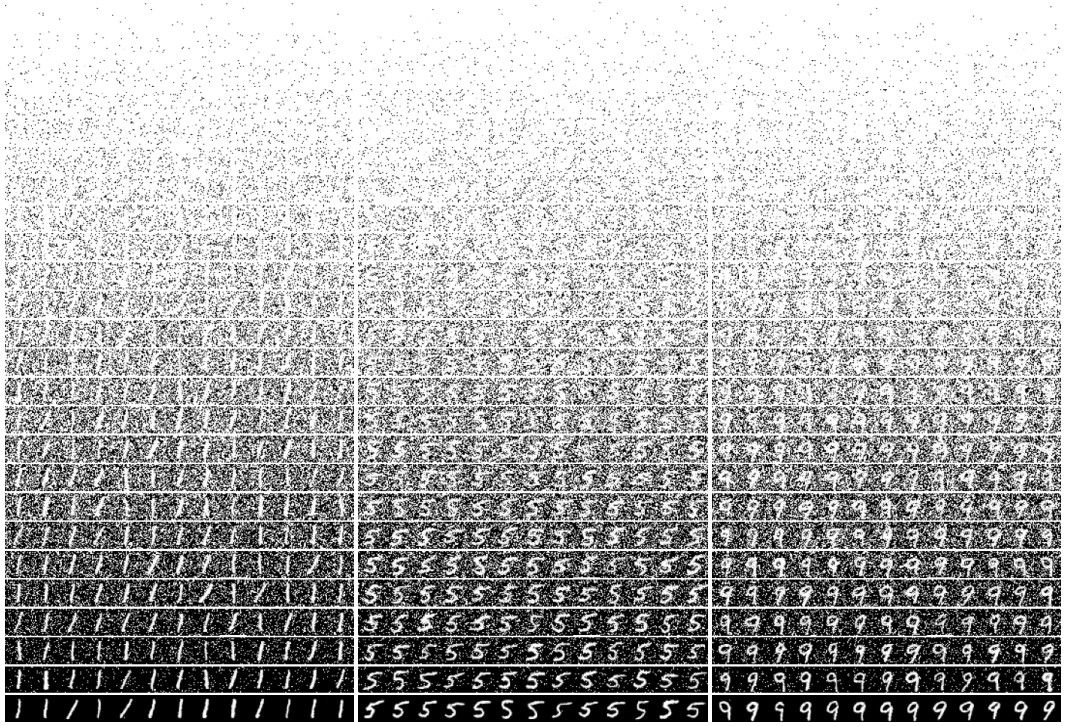

Figure 18: We visualise the PT trajectory along both the annealing path and PT iterations. For clarity, we thin the annealing path by a factor of 4 and record only every 8th PT iteration.

