# OpenReview forum: "CREPE: Controlling diffusion with REPlica Exchange"
_ICLR.cc/2026/Conference — ICLR 2026 Poster_

### Official Review · Reviewer_AFgB · 2025-11-01

**Soundness:** 1
**Presentation:** 3
**Contribution:** 2
**Rating:** 4
**Confidence:** 4

**Summary:**

This paper introduces CREPE (Controlling Diffusion with Replica Exchange), a framework for inference-time control of diffusion models via replica exchange (parallel tempering). The method adapts ideas from nonequilibrium statistical mechanics to generative inference by allowing multiple diffusion chains, each at different noise levels, to exchange information using accelerated parallel tempering (APT).

The authors derive swap acceptance ratios through Radon–Nikodym estimators between forward and backward path distributions, ensuring theoretically consistent swaps. Empirical evaluations on image and molecular generation tasks demonstrate that CREPE achieves comparable or slightly improved performance to SMC-based debiasing methods while enhancing sample diversity and supporting online refinement.

**Strengths:**

1. Applying replica exchange to diffusion inference is novel. It draws a meaningful connection between thermodynamic sampling methods and modern generative modeling.

2. The experiments cover several relevant applications—tempering, reward-tilting, model composition, and discrete diffusion.

3. Replica exchange could improve performance and sample diversity, which is well-studied in the MCMC area.

4. The paper is generally well-written, with clear motivation and good presentation.

**Weaknesses:**

My main concerns revolve around the practical overhead and complexity inherent in the Replica Exchange framework when applied to generative tasks, despite the theoretical benefits.

1. In my opinion, the advantages of replica exchange/parallel tempering are (1) potential performance improvement in some metrics and (2) increased sample diversity. However, the disadvantage of this method is also very clear: each sampling step requires a larger Number of Function Evaluations (NFEs). For instance, if you consider N chains, a standard step requires N steps + additional swap calculations (N+ NFEs), whereas a single-chain method only requires 1 NFE. While I understand that some replica exchange design in MCMC try to improve efficiency by optimizing chain swap rates and tuning various hyperparameters to outperform single-chain methods under the same NFEs, the problem remains significant: (1) You have a large number of hyperparameters that require fine-tuning, such as the temperature and step size for each chain, and the hyperparameters related to the swap design. This complexity is very challenging in generative tasks. I am not questioning the contribution of this paper, but rather, I do not believe that the replica exchange approach can be broadly effective in generative models. Although this method provides an elegant analytical framework, I do not believe it can both significantly improve performance and be simple to use for generative tasks.

2. MCMC-based sampling methods like CREPE necessitate a warm-up (burn-in samples) phase to achieve the target distribution, which requires potentially hundreds or even thousands of NFEs to reach low FIDs. This contrasts sharply with recent advancements in image generation, where distilled diffusion models [1-3] achieve competitive performance (e.g., FID below 2) on larger-resolution images using only a few NFEs or even one NFE. This fundamental requirement for a long burn-in period and high NFEs limits the applicability and competitive efficiency of MCMC sampling schemes like CREPE in real-time or resource-constrained engineering domains.

3. There is no discussion on the related work of replica exchange / parallel tempering. Since these techniques have been thoroughly studied in MCMC, it would strengthen the paper to discuss which insights from the MCMC community could transfer to generative settings (e.g., ladder design, replica spacing, or swap design).


4. The paper would benefit from diversity metrics (entropy, coverage, or mode count) to support this claim quantitatively, related to Figure 7 (replica exchange benefits sample diversity).

5. Too frequent swaps can hinder exploitation in low-temperature chains. The paper mentions the use of non-reversible even/odd communication but provides no statistics (e.g., acceptance rate, cycle time from the lowest to the highest temperature).

6. The swap acceptance rule appears to rely on exact energy or log-probability evaluations. It is unclear how this is handled under mini-batch training or high-dimensional data, where computing full log-likelihoods may be infeasible. Some clarification or approximation discussion would be valuable.

**References:**

[1] One-step Diffusion with Distribution Matching Distillation. CVPR 2024

[2] Consistency Trajectory Models: Learning Probability Flow ODE Trajectory of Diffusion. ICLR 2024

[3] Consistency Models Made Easy. ICLR 2025

**Questions:**

1. How sensitive is CREPE’s performance to the number of replicas and swap frequency?

2. What is the observed swap acceptance rate and average traversal time for a full temperature cycle in your experiments?

---

> ### Author Response · Authors · 2025-11-27
> **Reply to Reviewer AFgB (Part 1)**
>
> Thank you for your review. We are glad that you find our work novel, well-written, with clear motivation and good presentation. We address your concerns below:
>
> ---
>
> > ### My main concerns revolve around the practical overhead and complexity inherent in the Replica Exchange framework when applied to generative tasks, despite the theoretical benefits.
>
> We emphasise that these acceleration methods target a different goal: the works you mentioned are designed to speed up generation, whereas our work follows the line of inference-time control and scaling (e.g., TDS[1]/FKC[2]/FKS[3]/[4]), where one spends more compute at sampling time in order to steer the model towards some constraints.
>
> In this regime, a common solution is to use SMC. In our paper, we propose a new paradigm that uses replica exchange (parallel tempering). As we discuss in Appendix A.7 of the updated paper, PT and SMC have the same computational cost when generating the same number of particles. However, across our experiments, we empirically observe that PT offers several practical advantages over SMC.
>
>
>
> [1] Wu, Luhuan, et al. "Practical and asymptotically exact conditional sampling in diffusion models." NeurIPS.
>
> [2] Skreta, Marta, et al. "Feynman-kac correctors in diffusion: Annealing, guidance, and product of experts." ICML.
>
> [3] Singhal, Raghav, et al. "A general framework for inference-time scaling and steering of diffusion models." ICML.
>
>
> [4] Lee, Cheuk Kit, et al. "Debiasing guidance for discrete diffusion with sequential Monte Carlo." arXiv.
>
>
> > ### MCMC-based sampling methods like CREPE necessitate a warm-up (burn-in samples) phase to achieve the target distribution, which requires potentially hundreds or even thousands of NFEs to reach low FIDs.
>
> Burn-in / warm-up phase. We agree that, as with most MCMC-style methods, CREPE exhibits an initial burn-in phase. However, we believe this does not fundamentally limit its practicality in the regimes we target:
>
> (1) Exact-distribution tasks (e.g., molecular tempering).
> For tasks where the goal is to approximate an equilibrium distribution as accurately as possible (such as molecular tempering), we simply discard an initial portion of each chain as burn-in. In these settings, we typically generate a large ensemble of samples, so the cost of this initial phase is negligible and acceptable relative to the total NFE budget.
>
> (2) Controlled image generation (e.g., reward-tilting).
> For image-based control tasks, one may likewise discard early samples that do not yet satisfy the desired reward or constraint well. Even in this case, we find that samples obtained after burn-in exhibit large diversity. In contrast, SMC methods have no explicit burn-in, but under the same total NFE budget (including the burn-in NFEs of PT), they tend to collapse onto a small number of modes, so the effective number of distinct high-quality samples can be close to 1.
>
>
> (3) Reported Image experiments do not discard burn-in.
> Importantly, in our image experiments (Tab. 2), we do not discard burn-in samples: all samples from CREPE are included in the evaluation. Even under this stricter setting, CREPE still shows advantages over SMC once the sample size exceeds ~32.
>
> > ### There is no discussion on the related work of replica exchange / parallel tempering. Since these techniques have been thoroughly studied in MCMC, it would strengthen the paper to discuss which insights from the MCMC community could transfer to generative settings (e.g., ladder design, replica spacing, or swap design).
>
> Thank you for this suggestion. We provide a brief discussion in the conclusion and future works. We agree that these works can bring more insights to our setting. However, it is worth highlighting that, to the best of our knowledge, our work is essentially the first to connect relica exchange with controlling with generative settings in a rigorous way.

---

> > ### Author Response · Authors · 2025-11-27
> > **Reply to Reviewer AFgB (Part 2)**
> >
> > > ### Too frequent swaps can hinder exploitation in low-temperature chains. The paper mentions the use of non-reversible even/odd communication but provides no statistics.
> >
> >
> >
> > We respectfully disagree with the statement that "too frequent swaps can hinder exploitation in low-temperature chains." In replica exchange (PT), all swaps are Metropolis–Hastings corrected. As a result, changing how often we propose swaps does not bias the marginal distribution at each temperature: any accepted swap still leaves the low-temperature chain targeting the correct stationary distribution, and a swap does not reduce its ability to "exploit".
> >
> > Imagine the following thought experiment on a bimodal distribution with two well-separated modes. "Exploitation"  means spending time exploring one mode locally. If swaps frequently move the low-temperature chain between the two modes, then the algorithm simply alternates exploitation between the modes: whenever the replica is in a given mode, it still performs local, low-temperature updates there. Thus, exploitation is not lost; it is just taking place in different regions over time. The MH correction ensures this is correct.
> >
> >
> > Therefore, we do not see a principled argument that higher swap frequency inherently harms exploitation in low-temperature chains.
> >
> >
> >
> >
> > > ### The swap acceptance rule appears to rely on exact energy or log-probability evaluations. It is unclear how this is handled under mini-batch training or high-dimensional data, where computing full log-likelihoods may be infeasible. Some clarification or approximation discussion would be valuable.
> >
> > This is exactly what our work addresses. We have provided the explicit general form of the swap rate in Eq. 13 and include its detailed form for each task (tempering, composing, tilting) in Appendix A.5
> >
> > Intuitively, we do not need to know the log-probability for each PT level. Instead, we only need to know the ratio of log-probabilities between adjacent levels (See Eq. 12). The ratio can be approximated by the transition kernels of pretrained diffusion models (See Eq. 4). This allows us to calculate the swap rate without getting access to the exact marginal log-probability.
> >
> >
> > > ### Question 1: How sensitive is CREPE’s performance to the number of replicas and swap frequency?
> >
> > In CREPE, we use the noise levels defined by the diffusion model for the PT levels. In other words, the replica ladder is determined directly by the discretisation of the forward SDE/ODE. Empirically, we find that performance is fairly robust as long as adjacent noise levels are close enough that the swap acceptance rate is not too low -> 0.
> >
> >
> > We perform swaps at every PT iteration. This is, in fact, not a free hyperparameter (see Alg 1), so there is no sensitivity to tune with respect to it.
> >
> > > ### Question 2: What is the observed swap acceptance rate and average traversal time for a full temperature cycle in your experiments?
> >
> > Thank you for this question. Following your comment, we measured these quantities on the ALA tetrapeptide system. The swap acceptance rate ranges from approximately 0.3 to 0.7 across different time steps (PT levels), and a full temperature cycle (round trip) takes, on average, ~800 iterations. However, a full round trip is not required in our setting to obtain decorrelated samples: it suffices for a chain to traverse between the target level and a sufficiently high noise level. This is because CREPE moves toward higher noise, which is achieved simply by adding Gaussian noise, and a large enough noise makes the accepted sample ~independent from the starting sample. Considering this case, for example, a round trip between (t = 0) and (t = 1) takes on average ~275 iterations.

---

### Official Review · Reviewer_2CNP · 2025-11-01

**Soundness:** 3
**Presentation:** 3
**Contribution:** 2
**Rating:** 4
**Confidence:** 4

**Summary:**

The paper introduces CREPE, a new inference-time control method for diffusion models. Unlike common approaches based on Sequential Monte Carlo (SMC), CREPE sequentially generates samples using accelerated parallel tempering (APT; Zhang et al. 2025), a variant of parallel tempering MCMC that runs parallel chains at different diffusion timesteps and swaps samples between adjacent levels. The proposed method is empirically shown to approximate the target distribution more accurately and exhibits higher sample diversity compared to SMC-steering.

**Strengths:**

1. The paper is clearly written and easy to follow. Their derivations are mathematically rigorous.
2. The proposed algorithm is versatile and has broad practical potential for any application requiring inference-time control of diffusion models.
3. The experiments are promising and directly compare against strong SMC baselines. The results show that CREPE achieves competitive performance while maintaining higher sample diversity.

**Weaknesses:**

1. A major concern is that this work doesn't seem to provide methodological improvement over accelerated parallel tempering (APT). To me, it seems a direct application of APT to inference-time control with pretrained diffusion (with an assumption that the pretrained model perfectly models the time-reversal of a noising process). An algorithmic difference with APT, or an explanation on why applying APT to inference-time control is not trivial, would be helpful to understand the contribution.
2. Some details in the experiments section are missing, e.g., the number of random seeds used for each experiment, and the standard deviation for Table 2, runtime compared to SMC, etc. Also, the source code is not included.

I'm open to raising my score if those two points are resolved.

(Minor)
3. References need to be added in the appendix A.2., since some of them are studied in previous works.
4. Analysis of the acceptance probability would be valuable (e.g., how the choice of proposals affects the acceptance rates)
5. Typo, Line 198, the arrow for the backward process Markov
6. Type, Line 1088,"For the CREPE communication step." needs to be removed

**Questions:**

1. Line 96, "without explicit target densities": What does this mean? This claim does not seem valid for a reward-tilted setting.
2. How robust is the algorithm to the initialization, particularly when the number of iterations is limited?
3. Do SMC baselines use the same proposals as in Appendix A.2?
4. Do SMC baselines use standard techniques, like adaptive resampling (resampling when ESS below a certain threshold) and systematic sampling [1]?
5. Line 1040, "we run 50 batches of size 1000": Does this mean running SMC with 50,000 particles?

---

### References
[1] Douc, Randal, and Olivier Cappé. "Comparison of resampling schemes for particle filtering." ISPA 2005. Proceedings of the 4th International Symposium on Image and Signal Processing and Analysis, 2005. IEEE, 2005.

---

### LLM usage disclosure
I used LLM only to check grammar.

---

> ### Author Response · Authors · 2025-11-25
> **Reply to Reviewer 2CNP (Part 1)**
>
> Thank you for your thoughtful feedback. We address your questions below with released code and new experimental results:
>
> ---
>
> > ### A major concern is that this work doesn't seem to provide methodological improvement over accelerated parallel tempering (APT).
>
>
> Our goal is not to improve APT as a sampling algorithm. Rather, our contribution is to show how to use APT as a practical and theoretically grounded tool for inference-time control of diffusion models.
>
> > ### an explanation on why applying APT to inference-time control is not trivial, would be helpful to understand the contribution.
>
> APT is designed to draw samples from a given unnormalised density, assuming direct access to that density (up to a constant). In contrast, for CREPE, we do not have access to an explicit unnormalised density; instead, we only have a pretrained diffusion model.
>
> Moreover, most existing inference-time control methods for diffusion models are based on SMC. Designing an inference-time control algorithm that combines APT with diffusion dynamics, without explicit access to the target density and while still targeting the desired controlled distribution, is non-trivial. We will highlight this point in our newer version.
>
>
> > ### Some details in the experiments section are missing, ...  the source code is not included.
>
> Thank you for pointing out the missing details. For the molecular tempering experiments, we report results across 3 seeds, and for each run we evaluate a subset of 5,000 samples, since computing the W2 distance on all samples is prohibitively expensive. We will add standard deviations for the image experiments in the revised version. As discussed in Appendix A.7, the computational cost of SMC and CREPE is comparable. We did not report wall-clock runtime because we parallelise different SMC batches across separate jobs, which makes direct runtime comparison less meaningful.
>
> We provided our code below in the following link [https://anonymous.4open.science/r/crepe-code-BDAC/README.md]

---

> > ### Author Response · Authors · 2025-11-25
> > **Reply to Reviewer 2CNP (Part 2)**
> >
> > ### Questions:
> > > Line 96, "without explicit target densities": What does this mean? This claim does not seem valid for a reward-tilted setting.
> >
> > We mean that we do not have access to the full target density $\pi_0$. Even in the reward-tilted setting, we only know the value of the reward function evaluated on samples; we do not know the corresponding normalised or unnormalised density of the controlled diffusion distribution.
> >
> > > How robust is the algorithm to the initialization, particularly when the number of iterations is limited?
> >
> > We initialise the chains using the same heuristic proposal. This proposal is also the same as that used for the SMC baselines, to ensure a fair comparison.
> >
> > For the molecular experiments, we find that the choice of initialization is not critical in practice: the chains quickly converge to the target distribution. However, we agree that when the number of iterations is very limited, initialization can have a noticeable impact on performance. This is in fact analogous to the SMC setting: when the number of particles is small, the choice of proposal distribution becomes more important.
> >
> >
> > > Do SMC baselines use the same proposals as in Appendix A.2?
> >
> > Yes. For SMC with RNE, we use the same kernel as in Appendix A.2, and we have made this explicit in the revised version. For FKC, we use the process proposed in their work, which is in fact theoretically equivalent to the kernel of RNE we compare with (in continuous time).
> >
> > > Do SMC baselines use standard techniques, like adaptive resampling (resampling when ESS below a certain threshold) and systematic sampling [1]?
> >
> > We use systematic sampling and resample at every step for image experiments. This is because, for large models, the ESS at each step is typically very small.
> >
> > For molecule experiments, we resample every 4 steps in the main paper. We agree that adaptive resampling can be beneficial, and we have run additional experiments with adaptive resampling (resampling when ESS < 75%) and systematic sampling on Ala tetrapeptide and Ala hexapeptide. We include results for FKC (anneal score) and RNE. We omit FKC–anneal-noise here as it performs substantially worse than the other setups in our earlier results.
> >
> >
> > |  |  | FKC-anneal score | RNE | CREPE |
> > |:---:|:---:|:---:|:---:|:---:|
> > | Ala  Tetrapeptide  (800K -> 500K) | Energy TVD | **0.100   +- 0.006** | 0.141   +- 0.007 | 0.122 +- 0.004 |
> > |  | Distance TVD | **0.012   +- 0.001** | 0.020   +- 0.001 | **0.013+- 0.001** |
> > |  | Sample W2 | 0.911   +- 0.001 | 0.949   +- 0.003 | **0.856 +- 0.004** |
> > |  | TICA MMD | 0.119   +- 0.001 | 0.195   +- 0.009 | **0.035 +- 0.002** |
> > | Ala Hexapeptide  (800K -> 600K) | Energy TVD | **0.092   +- 0.005** | 0.070   +- 0.004 | 0.398 +- 0.001 |
> > |  | Distance TVD | 0.012   +- 0.001 | 0.013   +- 0.002 | **0.009 +- 0.001** |
> > |  | Sample W2 | 1.564   +- 0.004 | 1.556   +- 0.004 | **1.299 +- 0.004** |
> > |  | TICA MMD | 0.027   +- 0.001 | 0.077   +- 0.004 | **0.009 +- 0.001** |
> >
> >
> > We see that the SMC baselines still yield significantly worse Sample W2 and TICA MMD compared to CREPE, consistent with the trends reported in the main paper.
> >
> >
> > We also report the forward and reverse KL divergences for the TICA- samples between the model samples and the ground truth. The forward KL is particularly sensitive to mode coverage, and thus highlights how well each method captures the full support of the target distribution. As we can see, CREPE in fact achieves much better results compared to both FKC and RNE.
> >
> >
> > ##### **Forward-KL divergence**
> > |  | **FKC-Anneal Score** | **RNE** | **CREPE (ours)** |
> > |---|---|---|---|
> > | **ALA Tetrapeptide** | 26.712 | 26.436 | **9.965** |
> > | **ALA Hexapeptide** |9.793  | 9.556 | **4.721**  |
> >
> > ##### **Reverse-KL divergence**
> > |  | **FKC-Anneal Score** | **RNE** | **CREPE (ours)** |
> > |---|---|---|---|
> > | **ALA Tetrapeptide** | 1.284 | 1.114 | **0.451** |
> > | **ALA Hexapeptide** | 0.558 |  2.062 | **0.269** |
> >
> >
> > > Line 1040, "we run 50 batches of size 1000": Does this mean running SMC with 50,000 particles?
> >
> > This means repeat SMC with 1000 particles 50 times. It does not correspond to a single SMC run with 50,000 interacting particles, which is not practical.

---

### Official Review · Reviewer_LYC3 · 2025-11-01

**Soundness:** 2
**Presentation:** 3
**Contribution:** 3
**Rating:** 4
**Confidence:** 4

**Summary:**

This paper focuses on inference-time control of diffusion models. It proposes a method based on parallel tempering as an alternative to guidance and Sequential Monte Carlo (SMC) methods. The method is applied across annealing, reward tilting, model composition, and classifier-free guidance debiasing tasks.

**Strengths:**

1. The paper investigates a new direction of applying Parallel Tempering (PT; Replica Exchange MCMC), a widely used sampling algorithm, as an alternative to SMC for inference-time control of diffusion models.
2. The paper proposes a concrete algorithm tailored for diffusion models that can directly benefit from the properties of PT.
3. Experiments are conducted across diverse domains, including molecule sampling, image generation, and navigation, to demonstrate its effectiveness and versatility.

**Weaknesses:**

1. The limitations of SMC-based inference-time control methods, which are the key motivation of this paper, aren't convincing enough. It should be supported by more concrete statements or additional experiments. SMC is currently the most widely used method for inference-time control, and I hope the authors can provide more evidence on why the community should explore directions using PT. (See Question 1, 2, 3, 4)
2. The motivation of why PT is a promising alternative is not presented thoroughly in the introduction, and the explanation of why it could increase sample diversity isn't presented throughout the paper (or at least in the introduction or Section 3.3, where it should be).
3. Introduction claims that derivation of PT swap rates is their key contribution, but it's not presented in the main text.
4. It would be more interesting and convincing if there were applications of CREPE for more modern pre-trained models, such as the Stable Diffusion model family.

**Questions:**

1. Regarding Limitation 2 of SMC: It's confusing whether 'sample diversity' in line 77 refers to particle diversity during or at the end of SMC, or diversity of the sample distribution generated via SMC. In either case, it would be worth justifying the reason why it happens. Also, if the authors are referring to the first, one can run separate SMC samplers multiple times to sample diverse samples.
2. Related to Question 1, the claim that SMC requires a large batch of particles (Section 3.3, line 353-354) doesn't seem to hold for SMC-based inference-time control of diffusion, as recent works ([1], [2]) show that few as 4 particles are enough to sample high-quality samples. That said, running separate SMC samplers multiple times seems like a (computationally) viable strategy. Superiority of PT over SMC would be more convincing if the authors included this as a baseline.
3. Regarding Limitation 1 of SMC: SMC is memory-intensive due to saving particles. However, isn't CREPE also (perhaps even more) memory-intensive due to saving all timesteps? (This relates to the claim that SMC requires a large batch of particles in Question 2)
4. Regarding Limitation 3 of SMC: Indeed, basic SMC can't refine samples, but it can be naturally extended via Nested SMC where the outer sampling adds noise and denoise back, like how recent works ([3], [4]) do.

[1] Singhal, Raghav, et al. "A General Framework for Inference-time Scaling and Steering of Diffusion Models." Forty-second International Conference on Machine Learning.
[2] Kim, Sunwoo, Minkyu Kim, and Dongmin Park. "Test-time Alignment of Diffusion Models without Reward Over-optimization." The Thirteenth International Conference on Learning Representations.
[3] Uehara, Masatoshi, et al. "Reward-Guided Iterative Refinement in Diffusion Models at Test-Time with Applications to Protein and DNA Design." arXiv preprint arXiv:2502.14944 (2025).
[4] Ma, Nanye, et al. "Inference-time scaling for diffusion models beyond scaling denoising steps." arXiv preprint arXiv:2501.09732 (2025).

---

> ### Author Response · Authors · 2025-11-26
> **Reply to Reviewer LYC3 (Part 1)**
>
> Thank you for your review. We address your concerns and questions now. We see that most of your concerns are about our claim on the limitations of SMC. We therefore answer this first below.
>
> ---
>
>
>
> > ### Q1: regarding Limitation 2 of SMC: It's confusing whether 'sample diversity' in line 77 refers to particle diversity during or at the end of SMC, or diversity of the sample distribution generated via SMC. In either case, it would be worth justifying the reason why it happens. Also, if the authors are referring to the first, one can run separate SMC samplers multiple times to sample diverse samples.
>
> We mean both. The diversity at the end of SMC will also influence the diversity of the sample distribution generated via SMC.
>
> Indeed,  one can run separate SMC samplers multiple times to sample diverse samples. However, this will lead to high bias.  If we restart SMC many times, we may indeed obtain samples from different modes, but the relative mode weights now reflect the probability that a single SMC run collapses into each mode, rather than the true target mass of each mode. Thus, while the aggregated samples may appear diverse, the relative weights between modes are incorrect, and the resulting empirical distribution is systematically biased away from the true target.
>
> To show this bias, we have added a toy visualisation in our updated manuscript in Appendix A. 1.
>
>
> > ### Q2: related to Question 1, the claim that SMC requires a large batch of particles (Section 3.3, line 353-354) doesn't seem to hold for SMC-based inference-time control of diffusion, as recent works ([1], [2]) show that few as 4 particles are enough to sample high-quality samples. That said, running separate SMC samplers multiple times seems like a (computationally) viable strategy. Superiority of PT over SMC would be more convincing if the authors included this as a baseline.
>
>
> We agree that this can be a computationally feasible baseline. In fact, **when comparing with SMC (e.g., Tab1/2), we indeed compare with this setup**. Even in this case, CREPE presents obvious gains.
>
> However, we still want to highlight that SMC in this way can be systemtrically biased, as we discussed above and illustrated in Appendix A.1.
>
>
> > ### Q3: regarding Limitation 1 of SMC: SMC is memory-intensive due to saving particles. However, isn't CREPE also (perhaps even more) memory-intensive due to saving all timesteps? (This relates to the claim that SMC requires a large batch of particles in Question 2)
>
> No. CREPE only need to save the number of particles  = the number of PT levels in memory. This depends on the diffusion time steps. But it's typically 100-ish. However, for SMC, to ensure a valid distribution (like what we did for the Boltzmann distribution), we used 1000 samples per batch. This already considers the practicality! In fact, to do an SMC with a smaller bias, one might need 10,000 or even more batches.
>
>
> > ### Q4: Regarding Limitation 3 of SMC: Indeed, basic SMC can't refine samples, but it can be naturally extended via Nested SMC, where the outer sampling adds noise and denoise back, like how recent works ([3], [4]) do.
>
> We believe there is a misunderstanding. Whether this is correct depends critically on the choice of task. In most of the cases considered in the literature, this strategy is not theoretically justified.
>
> An implicit assumption for this strategy is that the intermediate targets satisfy:
> $$
> \pi_t = \pi_0 * \mathcal{N}
> $$
> where $\pi_t$ is the intermediate target corresponding to diffusion time $t$ and $\pi_0$ is the final target. **In other words, if one adds noise to the target at $t=0$, the noisy sample needs to follow the intermediate target $\pi_t$.  However, for tempering, composition, or general reward tilting, this assumption is typically wrong.** The only situation in which such a construction is theoretically justified is in reward tilting and the intermediate reward $r_t$ corresponds to the optimal value function --- a condition that does not hold in any of our experiments.
>
> Therefore, “adding noise repeatedly” may produce samples that look more diverse and can continue to refine samples, but this generally corresponds to a wrong approximation of the desired target distribution. Our method, in contrast, preserves the target (and intermediate) distributions, rather than relying on such heuristics.
>
>
> Additionally, even in the only case where this strategy is correct, nothing limits PT from taking the same strategy. One can view this strategy as adding a new outer loop beyond SMC/PT, which is orthogonal to what we propose.

---

> > ### Author Response · Authors · 2025-11-26
> > **Reply to Reviewer LYC3 (Part 2)**
> >
> > Now, we turn to address your concerns in weakness: We believe W1/2 have been addressed along with the questions.
> >
> > > Introduction claims that the derivation of PT swap rates is its key contribution, but it's not presented in the main text.
> >
> > We include this in Eq. 13 in the main and Appendix A.5 for details.
> >
> > We highlight that this formulation is non-trivial.  This is because both PT and APT are designed to draw samples from a given unnormalised density, assuming direct access to that density (up to a constant). In contrast, for CREPE, we do not have access to an explicit unnormalised density; instead, we only have a pretrained diffusion model.
> >
> >
> >
> > > It would be more interesting and convincing if there were applications of CREPE for more modern pre-trained models, such as the Stable Diffusion model family.
> >
> >
> >
> > Thank you for this suggestion. Our goal in this paper is to introduce CREPE as a **general inference-time control framework** and to demonstrate its applicability across **different modalities** (molecules, ImageNet-64/512, trajectories) and **tasks** (tempering, composition, stitching, reward-tilting). CREPE is architecture-agnostic and can be applied to Stable Diffusion in exactly the same way as to our current backbones. Therefore, we prioritised breadth and diversity of tasks. We agree that T2I experiments are an important direction which we will consider in our future work.

---

### Official Review · Reviewer_MkR6 · 2025-11-04

**Soundness:** 3
**Presentation:** 3
**Contribution:** 2
**Rating:** 4
**Confidence:** 3

**Summary:**

The paper proposes CREPE, an inference-time control framework for diffusion models based on replica exchange (accelerated parallel tempering), offering an alternative to SMC. Across molecules, images (ImageNet-64/512), and trajectory stitching, CREPE achieves competitive or better quality and diversity than SMC with similar NFE cost, while supporting online refinement and anytime sampling.

**Strengths:**

- Principled and general formulation.
- Practical advantages over SMC: better sample diversity after burn-in, lower memory (no large particle batches), online refinement and early stopping; compute parity analysis clarifies similar NFE cost.
- Broad, convincing experiments across modalities and tasks showing improved metrics (e.g., FID/IR, TICA diversity) and flexible compositions (e.g., CFG debiasing + reward tilting, trajectory stitching).

**Weaknesses:**

Novelty
- Important recent related works are missing [1,2,3]. Especially, DAS [1] avoids reward over-optimization by the tempering technique, thereby maintaining output diversity, and also deals with online reward optimization, which significantly affects the novelty of this work. The author should contrast how their proposed APT are better than DAS in terms of design.
- Some of the work uses sequential particle generation, rather than solely using parallel generation [3]


Experiment Setup
- Guidance results with recent T2I backbones (stable diffusions) are missing, which are widely used to compare the efficacy of inference-time guidance methods for diffusion [1,2,3].


---

[1] Test-time alignment of diffusion models without reward over-optimization, ICLR, 2025

[2] Inference-time scaling for diffusion models beyond scaling denoising steps, CVPR, 2025

[3] Inference-time Scaling of Diffusion Models through Classical Search, ArXiv, 2025

**Questions:**

- Why is generating particles sequentially better than generating them in parallel? Can you provide any theoretic background of this?

---

> ### Author Response · Authors · 2025-11-25
>
> Thank you for your feedback. We are glad you find our work principled and general with practical advantages and broad, convincing experiments. Below we address the concerns in detail.
>
> ---
> > ### Important recent related works are missing [1,2,3].
>
> Thank you for pointing out the related works. However, we respectfully disagree that these works reduce the novelty of CREPE. While they involve inference-time techniques for diffusion models, their goals, problem formulations, and sampling paradigms are fundamentally different from ours.
>
>
>
> #### **DAS [1]:**
>
> DAS [1] operates entirely within the SMC paradigm, with a specific focus on reward-tilting. It improves SMC by designing a better-aligned kernel. Importantly:
>
> - CREPE is not an SMC method. It is based on replica exchange rather than importance sampling + resampling. This is a new paradigm for inference-time control.
>
> - The goal is different. [1] focuses on reward-tilting, whereas our framework supports general tempering, composition, stitching, and reward tilting in a single unified framework.
>
> - CREPE can incorporate the improved kernels as well. Our formulation is orthogonal: any improved transition kernel designed for SMC can also be plugged into the proposal distribution used in CREPE. In our experiments, even with matched kernels, parallel tempering achieves significantly better diversity than SMC.
>
> Thus, DAS is fundamentally different from our contribution and it does not affect the novelty of introducing a parallel tempering paradigm for diffusion.
>
>
>
> #### **[2,3]:**
>
> [2] study inference-time scaling by searching over initial noises and analysing performance–compute trade-offs across different verifiers and search strategies. Their method does **not** target tempering, composition, or reward-tilting of the form we consider.
>
> [3] consider sampling from a target of the form  $\tilde p_0(x_0) \propto p_0(x_0) f(x_0)^\lambda$,  and propose a **classical search framework** combining global tree search (BFS/DFS-style) with local annealed Langevin MCMC. While this looks similar to our tilting target, their aim is to **bias sampling toward high-$f$ regions** instead of obtaining samples from the correct probabilistic distribution. Also, the approach does not provide a tempering and composition as in CREPE. For example, only by searching, it is unclear how to temper the distribution in a correct way.
>
>
> > ### Guidance results with recent T2I backbones (stable diffusions) are missing, which are widely used to compare the efficacy of inference-time guidance methods for diffusion [1,2,3].
>
> Our goal in this paper is to introduce CREPE as a **general inference-time control framework** and to demonstrate its applicability across **different modalities** (molecules, ImageNet-64/512, trajectories) and **tasks** (tempering, composition, stitching, reward-tilting). CREPE is architecture-agnostic and can be applied to Stable Diffusion in exactly the same way as to our current backbones. Therefore, we prioritised breadth and diversity of tasks over adding another T2I setup. We agree T2I experiments is an important direction for future work.
>
> > ### Why is generating particles sequentially better than generating them in parallel? Can you provide any theoretical background for this?
>
> This question really highlights the conceptual difference between SMC and parallel tempering (PT), rather than a simple "sequential vs. parallel" difference.
>
> In fact, what really matters is where one particle will dominate over other particles and lead to the collapse. Intuitively:
>
> - **SMC** simulates the diffusion time sequentially, propagating a batch of weighted particles in parallel. With a finite number of particles, if one particle happens to receive a much larger weight than the others, resampling can cause the system to collapse onto that particle.  To mitigate this, one needs to run SMC with a very large batch size, which can be very expensive to save in memory.
>
> - **PT** runs multiple noise levels in parallel and updates each chain sequentially via MCMC + swap moves. It is not based on importance weights and resampling, so it does not suffer from weight-degeneracy collapse. Swap rates can be poor if proposals are bad, but unlike SMC, this does not force all mass onto a single trajectory: when a proposal is accepted, we will be able to obtain a new sample.
> Also, because PT generates samples sequentially, we do not need to keep a large batch of particles in memory at once. The memory cost scales mainly with the number of replicas (PT levels), not with the total number of samples we ultimately generate.
>
>
>
> There is no paradigm that is universally superior. However, in the settings we study, SMC baselines tend to collapse onto a subset of modes and become very costly when increasing batch size, whereas CREPE maintains high diversity and more reliably realises the intended control.

---

### Author Response · Authors · 2025-11-25
**Clarifying Common Misconceptions About SMC and Its Connection to Parallel Tempering**

We thank all reviewers for their time and effort in evaluating our work. However, several of the comments reflect common conceptual confusions about how SMC behaves in the finite-particle regime. We clarify these points here, as they are central to interpreting our comparison between SMC and parallel tempering (PT), and may also be useful for future work.

---
> ### Misunderstandings 1: SMC can provide high-quality samples with very few particles (e.g., 4).

This is inaccurate. In general, SMC only enjoys **asymptotic guarantees**: accuracy is ensured only in the limit as the number of particles $N \to \infty$. With a very small number of particles, the particle system typically suffers from severe degeneracy and bias.

Some recent works report that SMC "works well" with very few particles, but this is often because their evaluation is not aligned with the actual target distribution. Concretely, some works consider reward-tilting targets of the form $p_0(x)\exp(r(x))$, but at evaluation time they only inspect image quality and the reward $r(x)$, rather than how well the samples match the true target $\pi_0(x)\propto p_0(x)\exp(r(x))$.

In such cases, SMC with four particles can indeed find samples with a higher reward $r(x)$ than a baseline without resampling, but the resulting empirical distribution can be very far from $\pi_0(x)$. In practice, this corresponds to a weak heuristic search for "better" samples, not principled SMC sampling from the correct tilted distribution.

By contrast, our method is explicitly designed to preserve the target distribution. This is clearly demonstrated in our tempering experiments, where we aim to obtain a sample from the correct distribution $\pi_0 \propto p^\beta$. In this regime, SMC with only 4 particles provides essentially no meaningful approximation to the true target.


> ### Misunderstandings 2: One can simply run multiple independent SMC samplers with small particle sets to obtain diverse samples.

This concern is closely related to the previous point. If one takes a large particle budget and splits it into many tiny independent SMC runs, the apparent diversity of individual samples may remain high, but the bias of each run is significantly increased.

To illustrate this issue, we update our paper with a toy experiment on a 1D GMM in Appendix A.1, where we compare SMC with mini-batches and CREPE in a single run. As shown, while mini-batch SMC can produce samples from all modes, the relative proportions of these modes are noticeably distorted, whereas CREPE preserves the true mode weights more faithfully. This is only a 1D toy example; in higher dimensions, samples from SMC in each batch will collapse further, and the misweighting becomes even more severe.

However, we also acknowledge that this mini-batch SMC heuristic can be acceptable in some application domains, especially when a strict distribution is not a priority. For this reason, our paper does include a comparison to such a setup on ImageNet. In this regime, parallel tempering (PT) still begins to show clear advantages once the particle size is moderate (e.g., around 32 particles; see Table 2). For the tempering application, we also compare our approach with the 500-particle SMC with 50 batches, and we can see that even if we use this mini-batch strategy, we still obtain samples with higher diversity.




> ### Misunderstandings 3: SMC can get higher diversity and  refined samples by adding noise repeatedly

Whether "adding noise repeatedly and re-denoising" is correct depends strongly on the task and on the structure of the intermediate targets. In most of the cases we considered, this strategy is not theoretically justified.

An implicit assumption for this strategy is that the intermediate targets satisfy:
$$
\pi_t = \pi_0 * \mathcal{N}
$$
where $\pi_t$ is the intermediate target corresponding to diffusion time $t$ and $\pi_0$ is the final target. In other words, if one adds noise to the target at $t=0$, the noisy sample needs to follow the intermediate target $\pi_t$.  However, for tempering, composition, or general reward tilting, this assumption is typically wrong. The only situation in which such a construction is theoretically justified is "reward tilting + the intermediate reward $r_t$ corresponds to the optimal value function" --- a condition that does not hold in any of our experiments.

Therefore, "adding noise repeatedly" may produce samples that look more diverse, but this generally corresponds to a wrong approximation of the desired target distribution. Our method, in contrast, preserves the target (and intermediate) distributions, rather than relying on such heuristics.


---
We thank the reviewer again for their time and effort in the reviewing process, and we hope this helps clarify some misunderstandings about our work and SMC.

---

### Meta-Review · Area_Chair_pkyb · 2025-12-23

**Summary:**

The paper proposes CREPE, a framework for inference-time control of diffusion models known as Replica Exchange. The authors present this method as a robust alternative to Sequential Monte Carlo (SMC), which is commonly used for steering diffusion outputs but often suffers from particle degeneracy and mode collapse. CREPE runs multiple diffusion chains in parallel at different noise levels (replicas) and performs swaps between them based on Radon-Nikodym estimators derived for implicit diffusion densities. The authors demonstrate the method's versatility across tasks such as molecular tempering, image reward-tilting, and trajectory stitching, claiming superior diversity and competitive performance compared to SMC.

The reviewers recognized the novelty and theoretical soundness of applying replica exchange to diffusion. Initial scores were unanimously 4, reflecting hesitation about the advantage over SMC and the omission of T2I baselines. AC’s recommendation of Accept is based on the authors' strong rebuttal, which clarified the theoretical advantages of PT regarding bias and diversity, and provided additional experiments comparing with SMC.

**Reviewer Concerns:**

Concerns Addressed by Rebuttal:
- Superiority of PT over SMC (Reviewers LYC3, MkR6, 2CNP): Multiple reviewers questioned why PT is better than SMC, especially given recent claims that SMC works with very few particles. The authors effectively addressed this by explaining that small-batch SMC is biased and fails to sample the true target distribution. They supported this with a new toy experiment (Appendix A.1) demonstrating that mini-batch SMC distorts mode weights, whereas CREPE preserves them.
- Novelty and Related Work (Reviewers MkR6, 2CNP): Reviewers pointed to other inference-time methods (e.g., DAS) as diminishing novelty. The authors successfully clarified that CREPE establishes a new sampling paradigm based on PT, distinct from the search or SMC-based optimization methods cited. They also clarified that adapting Accelerated PT (APT) to diffusion is non-trivial due to the lack of explicit densities.
- Practicality and Cost (Reviewers AFgB, MkR6): Concerns about high computational cost (NFE) and memory were addressed by clarifying that CREPE and SMC have comparable costs for the same particle budget. The authors also argued that burn-in is an acceptable trade-off for higher diversity in control tasks.
- Reproducibility (Reviewer 2CNP): The lack of code and experimental details was addressed by releasing the source code and adding standard deviations/seed details in the revision.


Concerns Outstanding:
- Modern Baselines (Reviewers MkR6, LYC3): The authors declined to add experiments with Stable Diffusion or other state-of-the-art T2I models, arguing their framework is architecture-agnostic. While methodologically valid, the absence of these popular baselines remains a minor weakness for impactful visual demonstration.

**Reviewer Scores:**

Reviewer MkR6 (4 to 5) questioned the theoretical advantage of sequential (PT) vs. parallel (SMC) generation. The authors' clarification about weight degeneracy in SMC and memory efficiency likely resolves this conceptual hurdle.

Reviewer LYC3 (4 to 5) was skeptical about the limitations of SMC. The new toy experiment demonstrating bias in mini-batch SMC provides the concrete evidence requested, though the reviewer may still prefer more modern baselines.

Reviewer 2CNP: (4 to 5) explicitly stated they were "open to raising [their] score" if code and missing details were provided. Since the authors released the code and added the requested statistical details, a score increase is highly probable.

Reviewer AFgB: (4 to 5). The concerns about practical overhead and burn-in were contextualized effectively (i.e., comparing to inference-time control baselines rather than fast distillations). The reviewer likely retains some reservations about complexity but should acknowledge the validity of the method.

---

### Decision · Program_Chairs · 2026-01-26

Accept (Poster)